# MODEL-BASED CAUSAL BAYESIAN OPTIMIZATION

**Scott Sussex**
ETH Zürich
scott.sussex@inf.ethz.ch

**Anastasiia Makarova**
ETH Zürich

**Andreas Krause**
ETH Zürich

## ABSTRACT

How should we intervene on an unknown structural causal model to maximize a downstream variable of interest? This optimization of the output of a system of interconnected variables, also known as causal Bayesian optimization (CBO), has important applications in medicine, ecology, and manufacturing. Standard Bayesian optimization algorithms fail to effectively leverage the underlying causal structure. Existing CBO approaches assume noiseless measurements and do not come with guarantees. We propose *model-based causal Bayesian optimization (*MCBO*)*, an algorithm that learns a full system model instead of only modeling intervention-reward pairs. MCBO propagates epistemic uncertainty about the causal mechanisms through the graph and trades off exploration and exploitation via the optimism principle. We bound its cumulative regret, and obtain the first non-asymptotic bounds for CBO. Unlike in standard Bayesian optimization, our acquisition function cannot be evaluated in closed form, so we show how the reparameterization trick can be used to apply gradient-based optimizers. Empirically we find that MCBO compares favorably with existing state-of-the-art approaches.

## 1 INTRODUCTION

Many applications, such as drug and material discovery, robotics, agriculture, and automated machine learning, require optimizing an unknown function that is expensive to evaluate. *Bayesian optimization (BO)* is an efficient framework for sequential optimization of such objectives (Močkus, 1975). The key idea in BO is to quantify uncertainty in the unknown function via a probabilistic model, and then use this to navigate a trade-off between selecting inputs where the function output is favourable (exploitation) and selecting inputs to learn more about the function in areas of uncertainty (exploration). While most standard BO methods focus on a black-box setup (Figure 1 a), in practice, we often have more structure on the unknown function that can be used to improve sample efficiency.

In this paper, we *exploit structural knowledge in the form of a causal graph* specified by a directed acyclic graph (DAG). In particular, we assume that actions can be modeled as interventions on a structural causal model (SCM) (Pearl, 2009) that contains the reward (function output) as a variable (Figure 1 b). While we assume the graph structure to be known, we consider the *functional relations* in the SCM as unknown. All variables in the SCM are observed along with the reward after each action. This Causal BO setting has important potential applications, such as optimizing medical and ecological interventions (Aglietti et al., 2020b). For illustrative purposes, consider the example of an agronomist trying to find the optimal Nitrogen fertilizer schedule for maximizing crop yield, described in Figure 1. There, the concentration of Nitrogen in the soil causes its concentration in the soil at the later timesteps.

To exploit the causal graph structure for optimization, we propose *model-based causal Bayesian optimization* (MCBO). MCBO explicitly models the full SCM and the accompanying uncertainty of all SCM components. This allows our algorithm to select interventions based on an optimistic strategy similar to that used by the upper confidence bound algorithm (Srinivas et al., 2010). We show that this strategy leads to the first CBO algorithm with a cumulative regret guarantee. For a practical algorithm, maximizing the upper confidence bound in our setting is computationally more difficult, because uncertainty in all system components must be propagated through the entire estimated SCM to the reward variable. We show that an application of the reparameterization trick allows MCBO to be practically implemented with common gradient-based optimizers. Empirically,

**Bayesian Optimization**

**Causal Bayesian Optimization**

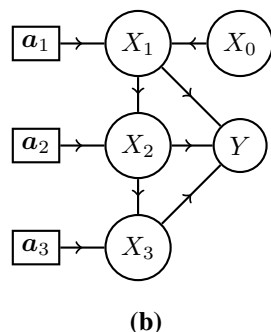

(a)

(b)

Figure 1: A visual comparison between the modelling assumptions of BO vs CBO. Circular nodes represent observed variables, squares represent action inputs and $Y$ is the reward. Algorithms select $\boldsymbol{a}$ before observing $\boldsymbol{X}$ and $Y$. **(a)** In standard BO, the DAG has the structure shown regardless of the problem structure. **(b)** The DAG corresponding to our stylised agronomy example, where we aim to maximize crop yield $Y$. CBO takes this DAG as input for designing actions. $X_0$ is an unmodifiable starting property of the soil, and $X_1 \ldots X_3$ are the measured amounts of Nitrogen in the soil at different timesteps. Each observation is modelled with its own Gaussian process. $\boldsymbol{a}_1 \ldots \boldsymbol{a}_3$ are possible interventions involving adding Nitrogen fertilizer to the soil.

MCBO achieves competitive performance on existing CBO benchmarks and a related setting called function network BO (Astudillo & Frazier, 2021b).

**Contributions**

- We introduce MCBO, a model-based algorithm for causal Bayesian optimization than can be applied with very generic classes of interventions.

- Using MCBO we prove the first sublinear cumulative regret bound for CBO. We show how the bound scales depending on the graph structure. We demonstrate that CBO can lead to a potentially *exponential* improvement in cumulative regret, with respect to the number of actions, compared to standard BO.

- By an application of the reparameterization technique, we show how our algorithm can be efficiently implemented with popular gradient-based optimizers.

- We evaluate MCBO on existing CBO benchmarks and the related setting of function network BO. Our results show that MCBO performs favorably compared to methods designed specifically for these tasks.

## 2 BACKGROUND AND PROBLEM STATEMENT

We consider the problem of an agent interacting with an SCM for $T$ rounds in order to maximize the value of a particular target variable. We start with introducing SCMs and the kinds of interventions an agent can perform on an SCM. In the following, we denote with $[m]$ the set of integers $\{0, \ldots, m\}$.

**Structural Causal Models**   An SCM is described by a tuple $\langle \mathcal{G}, Y, \boldsymbol{X}, \boldsymbol{F}, \boldsymbol{\Omega} \rangle$ of the following elements: $\mathcal{G}$ is a known DAG; $Y$ is the reward variable of interest; $\boldsymbol{X} = \{X_i\}_{i=0}^{m-1}$ is a set of observed random variables; the set $\boldsymbol{F} = \{\boldsymbol{f}_i\}_{i=0}^{m}$ defines the functional relations between these variables; and $\boldsymbol{\Omega} = \{\boldsymbol{\Omega}_i\}_{i=0}^{m}$ is a set of independent noise variables with zero-mean and known distribution.

We use the notation $Y$ and $X_m$ interchangeably and assume the elements of $\boldsymbol{X}$ are topologically ordered, i.e., $X_0$ is a root and $X_m$ is a leaf. We use the notation $pa_i \subset \{0, \ldots, m\}$ for the indices of the parents of the $i$th node, and $\boldsymbol{Z}_i = \{X_j\}_{j \in pa_i}$ for the parents of the $i$th node. We sometimes use $X_i$ to refer to both the $i$th node and the $i$th random variable.

Each $X_i$ is generated according to the function $\boldsymbol{f}_i : \mathcal{Z}_i \to \mathcal{X}_i$, taking the parent nodes $\boldsymbol{Z}_i$ of $X_i$ as input: $\boldsymbol{x}_i = \boldsymbol{f}_i(\boldsymbol{z}_i) + \boldsymbol{\omega}_i$, where lowercase denotes a realization of the corresponding random variable. The reward is a scalar $x_m \in \mathbb{R}$. An observation $X_i$ is defined over a compact set $\boldsymbol{x}_i \in \mathcal{X}_i \subset \mathbb{R}^d$, and its parents are defined over $\mathcal{Z}_i = \prod_{j \in pa_i} \mathcal{X}_j$ for $i \in [m-1]$.

**Interventions**   At each interaction round, the agent performs an *intervention* on the SCM. In this work, we consider two types of intervention models.

We consider a soft intervention model (Eberhardt & Scheines, 2007) where interventions are parameterized by controllable *action variables*. Let $\mathcal{A}_i \subset \mathbb{R}^q$ denote the compact space of an action $\boldsymbol{a_i}$ and $\mathcal{A}$ be the space of all actions $\boldsymbol{a} = \{\boldsymbol{a}_i\}_{i=0}^m$. We represent the actions as additional nodes in $\mathcal{G}$ (see Fig. 1): $\boldsymbol{a}_i$ is a parent of only $X_i$, and hence an additional input to $\boldsymbol{f}_i$. Since $\boldsymbol{f}_i$ is unknown, in our soft intervention model, the agent does not know apriori the functional effect of $\boldsymbol{a}_i$ on $\boldsymbol{X}_i$. A simple example of a soft intervention is a shift intervention $\boldsymbol{x}_i = \boldsymbol{f}_i(\boldsymbol{z}_i, \boldsymbol{a}_i) + \boldsymbol{\omega}_i = g_i(\boldsymbol{z}_i) + \boldsymbol{a}_i + \boldsymbol{\omega}_i$ for some function $g_i$. A shift intervention might occur in our example of adding Nitrogen fertilizer to soil and then measuring the total soil Nitrogen concentration.

While our theoretical results will focus on data obtained via soft interventions, our experiments also consider two other data sources. First, we consider a hard intervention model: hard interventions (often referred to as *do*-interventions) modify the targeted variable to a specific distribution independently of the variable's parents. For example, a doctor sets the dosage of a patient's medication, which fixes the dosage to a specific value (Aglietti et al., 2020b). Second, a special case under both intervention models is the collection of observational data, which is when no intervention is performed on the system. In the soft intervention model, not intervening on node $i$ is equivalent to setting $\boldsymbol{a}_i = \boldsymbol{0}$. An example would be not applying any Nitrogen fertilizer to the soil. In practice, the agent may have access to some previous observational data before its first interaction with the system. In the following, we introduce the problem setup under the soft intervention model and then adapt it to the hard intervention model.

**Constraints on interventions**   In many applications, we may not be able to intervene on all nodes simultaneously. For example, a farmer may only have the capacity to apply fertilizer at two out of three possible time windows in Fig. 1. This results in an action space with cardinality constraints, written as

$$\mathcal{A} = \left\{ \boldsymbol{a} = \{\boldsymbol{a}_i\}_{i=0}^m : \sum_{i=0}^m \mathbf{1}_{[\boldsymbol{a}_i \neq \boldsymbol{0}]} \leq c, c \geq 1 \right\}. \tag{1}$$

**Problem statement**   We consider the problem of an agent sequentially interacting with an SCM, with known DAG $\mathcal{G}$ and a fixed but unknown set of functions $\boldsymbol{F} = \{\boldsymbol{f}_i\}_{i=1}^m$ with $\boldsymbol{f}_i : \mathcal{Z}_i \times \mathcal{A}_i \to \mathcal{X}_i$. At round $t$ we select actions $\boldsymbol{a}_{:,t} = \{\boldsymbol{a}_{i,t}\}_{i=0}^m$ and obtain observations $\boldsymbol{x}_{:,t} = \{\boldsymbol{x}_{i,t}\}_{i=0}^m$, where we add an additional subscript to denote the round of interaction. The action $\boldsymbol{a}_{i,t}$ and the observation $\boldsymbol{x}_{i,t}$ are related by

$$\boldsymbol{x}_{i,t} = \boldsymbol{f}_i(\boldsymbol{z}_{i,t}, \boldsymbol{a}_{i,t}) + \boldsymbol{\omega}_{i,t}, \ \ \forall i \in [m]. \tag{2}$$

If $i$ corresponds to a root node, the parent vector $\boldsymbol{z}_{i,t}$ denotes an empty vector, and the output of $\boldsymbol{f}_i$ only depends on the action $\boldsymbol{a}_{i,t}$. Since we cannot intervene on the target variable $X_m$, we fix $\boldsymbol{a}_m = \boldsymbol{0}$. The reward is given by

$$y_t = f_m(\boldsymbol{z}_{m,t}, \boldsymbol{a}_{m,t}) + \boldsymbol{\omega}_{m,t}, \tag{3}$$

which implicitly depends on the whole intervention vector $\boldsymbol{a}_{:,t}$. We define the action that maximizes the expected reward by

$$\boldsymbol{a}^* = \arg\max_{\boldsymbol{a} \in \mathcal{A}} \mathbb{E}[y|\boldsymbol{a}], \tag{4}$$

where, unless otherwise stated, expectations are taken over noise $\boldsymbol{\omega}$.

**Performance metric**   Our agent's goal is to design a sequence of interventions $\{\boldsymbol{a}_{:,t}\}_{t=0}^T$ that achieves a high average expected reward. We hence study the *cumulative regret* (Lattimore & Szepesvári, 2020) over a time horizon $T$:

$$R_T = \sum_{t=1}^T \left[ \mathbb{E}[y|\boldsymbol{a}^*] - \mathbb{E}[y|\boldsymbol{a}_{:,t}] \right]. \tag{5}$$

A sublinear growth rate of $R_T$ with $T$ implies vanishing average regret: $R_T/T \to 0$ as $T \to \infty$. As an alternative to cumulative regret, one can also study the simple regret $\mathbb{E}[y|\boldsymbol{a}^*] - \mathbb{E}[y|\boldsymbol{a}_T]$. The most appropriate metric depends on the application. In the Nitrogen fertilizer example, cumulative regret might be preferable because we care about obtaining high crop yields across all years, not just in one final year.

**Index notation** Let $\boldsymbol{x}_{i,t} = [\boldsymbol{x}_{i,t,1}, \ldots, \boldsymbol{x}_{i,t,d}]^T$ denote a vector where $\boldsymbol{x}_{i,t,l}$ indicates indexing the component $l \in [d]$ of the $t$th timepoint of the observations at the node $i \in [m]$. For functions with vector output, e.g., $\boldsymbol{f}_i : \mathcal{Z}_i \to \mathcal{X}_i$, we sometimes consider notation with additional input to $\boldsymbol{f}_i$ that indicates the output dimension: $\boldsymbol{f}_i(\boldsymbol{z}, \boldsymbol{a}) = [f_i(\boldsymbol{z}, \boldsymbol{a}, 1), \ldots, f_i(\boldsymbol{z}, \boldsymbol{a}, d)]^T$.

**Regularity assumptions** We consider standard smoothness assumptions for the unknown functions $\boldsymbol{f}_i : \mathcal{S} \to \mathcal{X}_i$ defined over a compact domain $\mathcal{S}$ (Srinivas et al., 2010). In particular, for each node $i \in [m]$, we assume that $\boldsymbol{f}_i(\cdot)$ belongs to a reproducing kernel Hilbert space (RKHS) $\mathcal{H}_{k_i}$, a space of smooth functions defined on $\mathcal{S} = \mathcal{Z}_i \times \mathcal{A}_i$. This means that $f_i \in \mathcal{H}_{k_i}$ is induced by a kernel function $k_i : \tilde{\mathcal{S}} \times \tilde{\mathcal{S}} \to \mathbb{R}$ where $\tilde{\mathcal{S}} = \mathcal{S} \times [d]^1$. We also assume that $k_i(s, s') \leq \mathbf{1}$ for every $s, s' \in \tilde{\mathcal{S}}^2$. Moreover, the RKHS norm of $\boldsymbol{f}_i(\cdot)$ is assumed to be bounded $\|\boldsymbol{f}_i\|_{k_i} \leq \mathcal{B}_i$ for some fixed constant $\mathcal{B}_i > 0$. Finally, to ensure the compactness of the domains $\mathcal{Z}_i$, we assume that the noise $\boldsymbol{\omega}$ is bounded, i.e., $\boldsymbol{\omega}_i \in [-1, 1]^d$.[3]

**Problem statement under hard interventions** Under a hard intervention model, instead of selecting an action $\boldsymbol{a}$ in Eq. (4), the agent must select both a set of intervention targets $I \in \mathcal{I} \subset \mathcal{P}([m-1])$ and their values $\boldsymbol{a}_I \in \mathcal{A}_I \in \mathcal{A}$. For hard intervention we can rewrite Eq. (2) as

$$\boldsymbol{x}_i = \begin{cases} \boldsymbol{a}_i & \text{if } i \in I \\ \boldsymbol{f}_i(\boldsymbol{z}_i) + \boldsymbol{\omega}_i & \text{otherwise,} \end{cases} \qquad \forall \, i \in [m], \tag{6}$$

where $\boldsymbol{f}_i$ is unknown and employs the same regularity assumptions. Further constraints similar to Eq. (1) can be placed on either the intervention nodes $I$ or action values $\boldsymbol{a}_I$. Finally, observational data corresponds to the empty intervention set $I = \emptyset$.

## 2.1 RELATED WORK

Optimal decision-making in SCMs has been the subject of several recent works, for example, in the bandit setting (Lattimore et al., 2016; Bareinboim et al., 2015). Aglietti et al. (2020b) introduce causal BO (CBO) focusing on hard interventions. CBO considers unobserved confounding and uses the do-calculus to estimate $Y$ given $I, \boldsymbol{a}_I$ using both observational data and interventional data with the same intervention targets $I$. Aglietti et al. (2020a) extend CBO to make use of data obtained from hard interventions with different intervention targets. While both methods use do-calculus to estimate causal effects, they do not learn the full system model. Branchini et al. (2022) and Alabed & Yoneki (2022) extend these works to explore the CBO setting in the case of unknown DAG.

Function network BO (FNBO) (Astudillo & Frazier, 2021b) is similar to the CBO setup with soft interventions and the proposed algorithm uses an expected improvement acquisition function to select actions. MCBO generalizes the FNBO setup to a richer class of problems since the causal model formalism allows for modelling hard interventions. Moreover, in contrast to MCBO, FNBO assumes the system is noiseless which might be a restrictive assumption in practice. Kusakawa et al. (2021) study stochastic function networks in the special case of a chain graph. Similar to MCBO, they develop a UCB-based acquisition function and provide an accompanying cumulative regret guarantee. However, they do not employ a reparameterization trick to show how to optimize the acquisition function. Kusakawa et al. (2021) do not perform any empirical study of the proposed UCB-based method, but evaluate expected improvement and credible interval methods

---

[1] For vector-valued functions coming from an RKHS we consider a scalar-valued function where the output index is part of the function input, as described in Curi et al. (2020) Appendix F.

[2] This is known as the bounded variance property, and it holds for many common kernels.

[3] This assumption can be relaxed to $\boldsymbol{\omega}_i$ being sub-Gaussian using similar techniques to Curi et al. (2020) Appendix I. Though sub-Gaussian noise includes distributions with unbounded support, Curi et al. (2020) provide high probability bounds on the domain of $\mathcal{Z}_i$.

also developed for the chain graph setting. Both FNBO and CBO are part of a wider research direction on using intermediate observations from the computation of the unknown function to improve the sample efficiency of BO algorithms (Astudillo & Frazier, 2021a).

In concurrent work, Varici et al. (2022) consider a similar setup and prove a cumulative regret guarantee for a UCB-based algorithm with soft interventions. However, they focus only on linear SCMs and a binary action space for each intervention target, while MCBO applies to non-linear SCMs and a continuous action space.

The function class studied in this paper is similar to that of deep Gaussian processes (GPs) (Damianou & Lawrence, 2013) in that MCBO models $Y$ as a composition of GPs given $\boldsymbol{a}$. Deep GPs, however, do not make use of intermediate system variables and do not compose GPs according to a causal graph structure.

Our use of the reparameterization trick to practically implement an upper confidence bound acquisition function (Srinivas et al., 2010) in MCBO is inspired by Curi et al. (2020), who apply ideas from BO to design an algorithm for sample efficient reinforcement learning.

## 3 ALGORITHM

In this section, we propose the MCBO algorithm, describing the probabilistic model and acquisition function used. We first introduce MCBO under the soft intervention setup and then describe how to adapt it to hard interventions.

**Statistical model** We use Gaussian processes (GPs) for learning the RKHS functions $\boldsymbol{f}_0, \ldots, \boldsymbol{f}_m$ from observations. Our regularity assumptions permit the construction of confidence bounds using these GP models with priors associated with the RKHS kernels. We refer to Rasmussen (2003) for more background on the relation between GPs and RKHS functions. For all $i \in [m]$, let $\boldsymbol{\mu}_{i,0}$ and $\boldsymbol{\sigma}_{i,0}^2$ denote the prior mean and variance functions for each $\boldsymbol{f}_i$, respectively. Since $\boldsymbol{\omega}_i$ is bounded, it is also subgaussian and we denote the variance by $\rho_i^2$. The corresponding posterior GP mean and variance, denoted by $\boldsymbol{\mu}_{i,t}$ and $\boldsymbol{\sigma}_{i,t}^2$ respectively, are computed based on the previous evaluations $\mathcal{D}_t = \{\boldsymbol{z}_{:,1:t}, \boldsymbol{a}_{:,1:t}, \boldsymbol{x}_{:,1:t}\}$. In particular, for each function $f_i(\cdot, \cdot, l)$ defined by the given kernel $k_i$ and output component $l$:

$$\mu_{i,t}(\boldsymbol{z}_i, \boldsymbol{a}_i, l) = \boldsymbol{k}_{i,t}(\boldsymbol{z}_i, \boldsymbol{a}_i, l)^\top (\boldsymbol{K}_t + \rho_i^2 \boldsymbol{I})^{-1} \mathrm{vec}(\boldsymbol{x}_{i,1:t}) , \tag{7}$$

$$\sigma_{i,t}^2(\boldsymbol{z}_i, \boldsymbol{a}_i, l) = k_i((\boldsymbol{z}_i, \boldsymbol{a}_i, l); (\boldsymbol{z}_i, \boldsymbol{a}_i, l)) - \boldsymbol{k}_{i,t}(\boldsymbol{z}_i, \boldsymbol{a}_i, l)^\top (\boldsymbol{K}_t + \rho_i^2 \boldsymbol{I})^{-1} \boldsymbol{k}_{i,t}(\boldsymbol{z}_i, \boldsymbol{a}_i, l) , \tag{8}$$

where $\boldsymbol{I}$ denotes the identity matrix, $\mathrm{vec}(\boldsymbol{x}_{i,1:t}) = [\boldsymbol{x}_{i,1,1}, \boldsymbol{x}_{i,1,2}, \ldots, \boldsymbol{x}_{i,t,d}]^\top$ and for $(t_1, l), (t_2, l') \in [(1,1), (1,2), \ldots, (t,d)]$:

$$[\boldsymbol{K}_t]_{(t_1,l),(t_2,l')} = k_i((\boldsymbol{z}_{i,t_1,l}, \boldsymbol{a}_{i,t_1,l}, l); (\boldsymbol{z}_{i,t_2,l'}, \boldsymbol{a}_{i,t_2,l'}, l')),$$

$$\boldsymbol{k}_{i,t}(\boldsymbol{z}_i, \boldsymbol{a}_i, l)^\top = [k_i((\boldsymbol{z}_{i,1,1}, \boldsymbol{a}_{i,1,1}, 1); (\boldsymbol{z}_i, \boldsymbol{a}_i, l)), \ldots, k_i((\boldsymbol{z}_{i,t,d}, \boldsymbol{a}_{i,t,d}, d); (\boldsymbol{z}_i, \boldsymbol{a}_i, l))]^\top.$$

We write $\boldsymbol{\mu}_{i,t} = [\mu_{i,t}(\cdot, 1), \ldots, \mu_{i,t}(\cdot, d)]^T$ and similarly for $\boldsymbol{\sigma}_{i,t}$. We give more background on the posterior updates of vector-valued GPs in Appendix A.1.

At time $t$, the known set $\mathcal{M}_t$ of statistically plausible functions $\tilde{\boldsymbol{F}} = \{\tilde{\boldsymbol{f}}_i\}_{i=0}^m$ (functions that lie inside the confidence interval given by the posterior of each GP) is defined as:

$$\mathcal{M}_t = \left\{ \tilde{\boldsymbol{F}} = \{\tilde{\boldsymbol{f}}_i\}_{i=0}^m, \text{ s.t. } \forall i : \tilde{\boldsymbol{f}}_i \in \mathcal{H}_{k_i}, \|\tilde{\boldsymbol{f}}_i\|_{k_i} \leq \mathcal{B}_i, \right.$$

$$\left. \text{and } \left| \tilde{\boldsymbol{f}}_i(\boldsymbol{z}_i, \boldsymbol{a}_i) - \boldsymbol{\mu}_{i,t-1}(\boldsymbol{z}_i, \boldsymbol{a}_i) \right| \leq \beta_{i,t} \boldsymbol{\sigma}_{i,t-1}(\boldsymbol{z}_i, \boldsymbol{a}_i), \forall \boldsymbol{z}_i \in \mathcal{Z}_i, \boldsymbol{a}_i \in \mathcal{A}_i \right\}. \tag{9}$$

Here, $\beta_{i,t}$ is a parameter that ensures the validity of the confidence bounds. Some examples of concentration inequalities under similar regularity assumptions as well as explicit forms for $\beta_{i,t}$ can be found in Chowdhury & Gopalan (2019) and Srinivas et al. (2010). In the following, we set $\beta_{i,t} = \beta_t$ for all $i$ such that the confidence bounds in Eq. (9) are still valid.

---

**Algorithm 1** Model-based Causal BO (MCBO)

---

**Require:** Parameters $\{\beta_t\}_{t \geq 1}, \mathcal{G}, \mathbf{\Omega}$, kernel functions $k_i$ and prior means $\boldsymbol{\mu}_{i,0} = \mathbf{0} \ \forall i \in [m]$
  1: **for** $t = 1, 2, \ldots$ **do**
  2:     Construct confidence bounds as in Eq. (9)
  3:     Select $\boldsymbol{a}_t \in \arg\max_{\boldsymbol{a} \in \mathcal{A}} \max_{\boldsymbol{\eta}(\cdot)} \mathbb{E}[y \,|\, \{\tilde{\boldsymbol{f}}\}, \boldsymbol{a}]$ as in Eq. (12)
  4:     Observe samples $\{\boldsymbol{z}_{i,t}, \boldsymbol{x}_{i,t}\}_{i=0}^m$
  5:     Use $\mathcal{D}_t$ to update posterior $\{\boldsymbol{\mu}_{i,t}(\cdot), \boldsymbol{\sigma}_{i,t}^2(\cdot)\}_{i=0}^m$ as in Eqs. (7) and (8)
  6: **end for**

---

**Algorithm 2** Model-based Causal BO with Hard Interventions (MCBO)

---

**Require:** Parameters $\{\beta_t\}_{t \geq 1}, \mathcal{G}, \mathbf{\Omega}$, kernel functions $k_i$ and prior means $\boldsymbol{\mu}_{i,0} = \mathbf{0} \ \forall i \in [m]$
  1: **for** $t = 1, 2, \ldots$ **do**
  2:     Construct confidence bounds as in Eq. (9)
  3:     Select $I, \boldsymbol{a}_I \in \arg\max_{I, \boldsymbol{a}_I} \max_{\boldsymbol{\eta}} \mathbb{E}[y \,|\, \{\tilde{\boldsymbol{f}}\}, do(X_I = \boldsymbol{a}_I)]$
  4:     Observe samples $\{\boldsymbol{z}_{i,t}, \boldsymbol{x}_{i,t}\}_{i=0}^m$
  5:     Use $\mathcal{D}_t$ to update posterior $\{\boldsymbol{\mu}_{i,t}(\cdot), \boldsymbol{\sigma}_{i,t}^2(\cdot)\}_{i=0}^m$ as in Eqs. (7) and (8)
  6: **end for**

---

**Acquisition function**   At each round $t$, interventions are selected based on maximizing an acquisition function. Our acquisition function is based on the upper confidence bound acquisition function (Srinivas et al., 2010). That is, we optimistically pick interventions that yield the highest expected return among all system models that are still plausible given past observations:

$$\boldsymbol{a}_{:,t} = \arg\max_{\boldsymbol{a} \in \mathcal{A}} \ \max_{\tilde{\boldsymbol{F}} \in \mathcal{M}_t} \ \mathbb{E}_{\boldsymbol{\omega}}\Big[y \,|\, \tilde{\boldsymbol{F}}, \boldsymbol{a}\Big]. \tag{10}$$

Note that Eq. (10) is not amenable to commonly used optimization techniques, due to the maximization over a set of functions with bounded RKHS norm. Therefore, following Curi et al. (2020), we use the reparameterization trick to write any $\tilde{\boldsymbol{f}}_i \in \tilde{\boldsymbol{F}} \in \mathcal{M}_t$ using a function $\boldsymbol{\eta}_i : \mathcal{Z}_i \times \mathcal{A}_i \to [-1, 1]^{d_i}$ as:

$$\tilde{\boldsymbol{f}}_{i,t}(\tilde{\boldsymbol{z}}_i, \tilde{\boldsymbol{a}}_i) = \boldsymbol{\mu}_{i,t-1}(\tilde{\boldsymbol{z}}_i, \tilde{\boldsymbol{a}}_i) + \beta_t \boldsymbol{\sigma}_{i,t-1}(\tilde{\boldsymbol{z}}_i, \tilde{\boldsymbol{a}}_i) \circ \boldsymbol{\eta}_i(\tilde{\boldsymbol{z}}_i, \tilde{\boldsymbol{a}}_i), \tag{11}$$

where $\tilde{\boldsymbol{x}}_i = \tilde{\boldsymbol{f}}_i(\tilde{\boldsymbol{z}}_i, \tilde{\boldsymbol{a}}_i) + \tilde{\boldsymbol{\omega}}_i$ denotes observations from simulating actions in one of the plausible models, and not necessarily the true model. $\circ$ denotes the elementwise multiplication of vectors. This reparametrization allows for rewriting our acquisition function in terms of $\boldsymbol{\eta} : \mathcal{Z} \times \mathcal{A} \to [-1, 1]^{|\mathcal{X}|}$:

$$\boldsymbol{a}_{:,t} = \arg\max_{\boldsymbol{a} \in \mathcal{A}} \max_{\boldsymbol{\eta}(\cdot)} \mathbb{E}_{\boldsymbol{\omega}}\Big[y \,|\, \tilde{\boldsymbol{F}}, \boldsymbol{a}\Big], \ \text{ s.t. } \tilde{\boldsymbol{F}} = \{\tilde{\boldsymbol{f}}_{i,t}\} \ \text{in Eq. (11)}. \tag{12}$$

Intuitively, the variables $\boldsymbol{\eta}$ allow for choosing optimistic but plausible models given the confidence bounds. In practice, the function $\boldsymbol{\eta}$ can be parameterized by, for example, a neural network, and then standard optimization techniques are applied. For the theory, we assume access to an oracle providing the global optimum for Eq. (12). In practice, such an oracle may be computationally infeasible due to the non-convexity of Eq. (12). We discuss heuristics that we use for approximating this oracle in Appendix A.3.

Algorithm 1 summarizes our Model-based Causal BO approach. We note that for the special case of the SCM following the DAG of Fig. 1(a), our algorithm and the associated guarantees reduce to standard BO (Srinivas et al., 2010).

**Hard interventions**   MCBO also naturally generalizes to hard interventions (Algorithm 2). In our experiments, we perform the combinatorial optimization over the set of nodes $\mathcal{I}$ by enumeration because $|\mathcal{I}|$ is not large for the instances we consider. We apply the notion of a minimal intervention set from Lee & Bareinboim (2019) to prune sets of intervention targets that contain redundant interventions, resulting in a smaller set to optimize over.

# 4 THEORETICAL ANALYSIS

This section describes the convergence guarantees for MCBO under a soft intervention model, showing the first sublinear cumulative regret bounds for causal BO. We start by introducing additional technical assumptions required for the analysis.

**Assumption 1** (*Functional relations*). All $\boldsymbol{f}_i \in \boldsymbol{F}$ are $L_f$-Lipschitz continuous.

**Assumption 2** (*Continuity*). $\forall i, t$, the functions $\boldsymbol{\mu}_{i,t}, \boldsymbol{\sigma}_{i,t}$ are $L_\mu, L_\sigma$ Lipschitz continuous.

**Assumption 3** (*Calibrated model*). All statistical models are *calibrated* w.r.t. $\boldsymbol{F}$, so that $\forall i, t$ there exists a sequence $\beta_t \in \mathbb{R}_{>0}$ such that, with probability at least $(1 - \delta)$, for all $\boldsymbol{z}_i, \boldsymbol{a}_i \in \mathcal{Z}_i \times \mathcal{A}_i$ we have $|\boldsymbol{f}_i(\boldsymbol{z}_i, \boldsymbol{a}_i) - \boldsymbol{\mu}_{i,t-1}(\boldsymbol{z}_i, \boldsymbol{a}_i)| \leq \beta_t \boldsymbol{\sigma}_{i,t-1}(\boldsymbol{z}_i, \boldsymbol{a}_i)$, element-wise.

Assumption 1 follows directly from the regularity assumptions of Section 2. Assumption 2 holds if the RKHS of each $\boldsymbol{f}_i$ has a Lipschitz continuous kernel (see Curi et al. (2020), Appendix G). Assumption 2 restricts the convergence guarantees to soft interventions that affect their target variable in a smooth way, meaning that our analysis does not directly apply to the hard intervention model. We nevertheless experimentally demonstrate the effectiveness of MCBO in non-smooth settings, such as CBO with hard interventions. Assumption 3 holds when we assume that the $i$th GP prior uses the same kernel as the RKHS of $\boldsymbol{f}_i$ and that $\beta_t$ is sufficiently large to ensure the confidence bounds in Eq. (9) hold.

In the DAG $\mathcal{G}$ over nodes $\{X_i\}_{i=0}^m$, let $N$ denote the maximum distance from a root node to $X_m$: $N = \max_i \text{dist}(X_i, X_m)$ where $\text{dist}(\cdot, \cdot)$ is measured as the number of edges in the longest path from a node $X_i$ to the reward node $X_m$. Let $K$ denote the maximum number of parents of any variable in $\mathcal{G}$: $K = \max_i |pa(i)|$. The following theorem bounds the performance of MCBO in terms of cumulative regret.

**Theorem 1.** *Consider the optimization problem in Eq.* (4) *with SCM satisfying Assumptions 1–3 where $\mathcal{G}$ is known but $\boldsymbol{F}$ is unknown. Then for all $T \geq 1$, with probability at least $1 - \delta$, the cumulative regret of Algorithm 1 is bounded by*

$$R_T \leq \mathcal{O}\left(L_f^N L_\sigma^N \beta_T^N K^N m \sqrt{T \gamma_T}\right).$$

Here, $\gamma_T = \max_i \gamma_{i,T}$ where the node-specific $\gamma_{i,T}$ denotes the *maximum information gain* at a time $T$ commonly used in the standard regret guarantees for Bayesian optimization (Srinivas et al., 2010). This maximum information gain is known to be sublinear in $T$ for many common kernels, such as linear and squared exponential kernels, resulting in an overall sublinear regret for MCBO. We refer to Appendix A.2.3 for the proof.

Theorem 1 demonstrates that the use of the graph structure in MCBO results in a potentially *exponential improvement* in how the cumulative regret scales with the number of actions $m$. Standard Bayesian optimization as in Fig. 1 (a), that makes no use of the graph structure, results in cumulative regret *exponential* in $m$ (Srinivas et al., 2010), when using a squared exponential kernel. When all $X_i$ in MCBO are modeled with squared exponential kernels that model output components independently, we have $\gamma_T = \mathcal{O}(d(Kd + q)(\log T)^{Kd+q+1})$, resulting in a cumulative regret that is exponential in $K$ and exponential in $N$. However, note that $m \geq K + N$. For several common graphs, the exponential scaling in $N$ and $K$ could be significantly more favorable than the exponential scaling in $m$. Consider the case of $\mathcal{G}$ having the binary-tree-like structure in appendix Fig. 3 (binary tree), where $N = \log(m)$ and $K = 2$. In such settings, the cumulative regret of MCBO will have only *polynomial* dependence on $m$. We further discuss the bound in Theorem 1 for specific kernels in Appendix A.2.3 and discuss the dependence of $\beta_T$ on $T$ in Appendix A.2.4.

# 5 EXPERIMENTS

In this section, we empirically evaluate MCBO on six problems taken from previous CBO or function network papers (Aglietti et al., 2020b; Astudillo & Frazier, 2021b). The DAGs corresponding to each task are given in Fig. 3 of the appendix. We provide an open-source implementation of MCBO[4].

---

[4]`https://github.com/ssethz/mcbo`

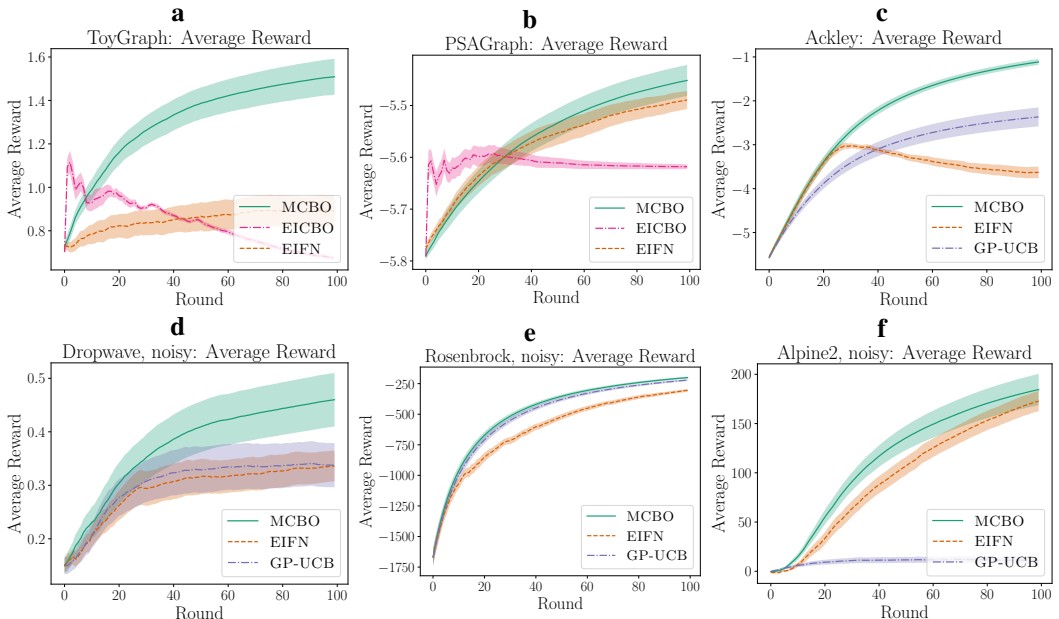

Figure 2: **(a, b)** EICBO does not achieve monotonically improving average reward on CBO tasks, while MCBO achieves high average reward on both tasks. **(c)** On the noiseless Ackley function networks task, EIFN also does not achieve monotonically improving average reward. **(d, e, f)** On a range of noisy function networks tasks, MCBO is at as good as and often higher performing than EIFN and GP-UCB baselines.

**Baselines** We compare our approach with three baselines (i) Expected Improvement for Function Networks (EIFN) (Astudillo & Frazier, 2021b); (ii) Causal Bayesian Optimization (EICBO) (Aglietti et al., 2020b); and (iii) standard upper confidence bound (GP-UCB) (Srinivas et al., 2010) which models the objective given interventions with a single GP (see Fig. 1 a). To enable a fair comparison, we only apply EICBO to the hard intervention setting, since it was designed specifically for a hard intervention model.

**Experimental setup** We report the average reward as a function of the number of system interventions performed. The average reward at time $T$ is defined by $\sum_{t=0}^{T} \mathbb{E}_{\omega}[Y \mid \boldsymbol{a}_t]$ and is directly inversely related to cumulative regret in that high average expected reward is equivalent to low cumulative regret. This matches the performance metric studied in our analysis. Average expected reward and cumulative regret are natural metrics for many real applications, like crop management, in which we want consistently high yield, and the healthcare-inspired setting we study in these experiments, where we seek good treatment outcomes for more than a single patient. In the appendix, we show experiments measuring the best expected reward of any action previously chosen, which is more similar to an inverse of the simple regret. We report mean performance over 20 random seeds, with error bars showing $\pm \sigma/\sqrt{20}$ where $\sigma$ is the standard deviation across the repeats. All figures that are referenced but not in the main paper can be found in the appendix.

For the guarantees in Theorem 1 to hold, $\{\beta_t\}_{t=0}^{T}$ must be chosen so that the model is calibrated at all time-steps as in Eq. (9). In practice, we select a single $\beta$ such that $\beta_t = \beta, \forall t$. Choosing $\beta$ too pessimistically will result in high regret, as demonstrated by the dependence of the guarantee on $\beta^N$. For GP-UCB and MCBO, $\beta$ is chosen by cross-validation across tasks, as described in the appendix.

**Toy Experiment** First, we evaluate on the synthetic ToyGraph setting from Aglietti et al. (2020b). ToyGraph is a hard intervention CBO task where $\mathcal{I} = \{\emptyset, \{0\}, \{1\}\}$. All methods start with 10 observational samples and samples from 2 random interventions on each $I \in \mathcal{I}$. When EICBO obtains interventional data, it obtains noiseless samples because it is not designed for the noisy setting. By noiseless samples, we mean that for action $\boldsymbol{a}$ EICBO observes $\mathbb{E}_{\omega}[Y \mid \boldsymbol{a}]$. Other methods obtain single samples from the distribution $Y, X \mid \boldsymbol{a}$.

In ToyGraph, $I = \{1\}$ is the optimal target, but to efficiently learn the optimal $a_{\{1\}}$, the agent must generalize from both observational data and interventional data on other targets $I = \{0\}$. EICBO models $Y$ given $a_I$ with a separate GP for every $I$ and is consequently only able to make use of observational data and interventional data with the same target. Since MCBO learns a full system model, it incorporates all observations into the learned model, even when interventions do not match.

Figure 2 (a) shows that the average reward of MCBO (Algorithm 2) is favorable compared to the baselines. Both baselines are built on expected improvement, which will continue to explore even after high-reward solutions are found. This explains the non-monotonic average reward of EICBO.

**Healthcare Experiment** PSAGraph is inspired by the DAG from a real healthcare setting (Ferro et al., 2015) and also benchmarked by Aglietti et al. (2020b). The agent intervenes by prescribing statins and/or aspirin while specifying the dosage to control prostate-specific antigen (PSA) levels. Here $\mathcal{I} = \{\emptyset, \{2\}, \{3\}, \{2, 3\}\}$ and all interventions are hard interventions. Initial sample sizes are the same as for ToyGraph. Figure 2 (b) again shows EICBO having a nonmonotonic average reward and strong comparable performance of MCBO.

**Noiseless Function Networks** In addition to the hard intervention setting, we evaluate MCBO and the baselines on four tasks from Astudillo & Frazier (2021b). All systems have up to six nodes and varying graph structures. In function networks, actions can affect multiple system variables, and system variables can be children of multiple actions. Function networks are deterministic, so $\omega = 0$. MCBO (Algorithm 1) can be applied directly in this setting, and the guarantees are also easily transferable. Like in Astudillo & Frazier (2021b), there are no constraints (besides bounded domain) on actions, and the agent is initialized with $2A + 1$ samples from random actions, where $A$ is the number of action nodes.

Figure 2 (c) and Figure 4 (a,b,c) show that MCBO achieves competitive average reward on all tasks. EIFN is better on Dropwave. Meanwhile, MCBO is substantially better than EIFN on the Ackley and Rosenbrock tasks. Overall, there are not sufficiently many tasks established in the literature to conclusively say which properties might make a task favor EIFN over MCBO. However, this would be interesting to understand in future work and likely relates to the wider conversation in BO comparing expected improvement and UCB algorithms (Merrill et al., 2021). We find that the naive GP-UCB approach, which does not use the graph structure, generally performs poorly, especially on problems with larger graphs like Alpine2. On Ackley, EIFN does not achieve monotonically improving average reward, which is not unexpected given that it is based upon expected improvement.

**Noisy Function Networks** We modify three of the function networks settings to include an additive zero-mean Gaussian noise at every system variable, making $\omega$ non-zero. EIFN is designed for deterministic function networks and has no convergence guarantees in this setting. Results in terms of average (Figure 2 d,e,f) and best reward (Figure 6 g, h, i) are comparable to the noiseless case, with MCBO and EIFN both performing well compared to GP-UCB.

## 6 CONCLUSION

This paper introduces MCBO, a principled model-based approach to solving Bayesian optimization problems over structural causal models. Our approach explicitly models all variables in the system and propagates epistemic uncertainty through the model to select interventions based on the optimism principle. This allows MCBO to solve global optimization tasks in systems that have known causal structure with improved sample efficiency compared to prior works. We prove the first non-asymptotic convergence guarantees for an algorithm solving the causal Bayesian optimization problem and demonstrate that its theoretical advantages are reflected in strong empirical performance. Future work might consider how to apply the method to large graphs, where the sets of all possible discrete intervention targets cannot be efficiently enumerated.

## ACKNOWLEDGMENTS AND DISCLOSURE OF FUNDING

We thank Lars Lorch, Parnian Kassraie and Ilnura Usmanova for their feedback and anonymous reviewers for their helpful comments. This research was supported by the Swiss National Science Foundation under NCCR Automation, grant agreement 51NF40 180545, and by the European Research Council (ERC) under the European Union's Horizon grant 815943.

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

# A APPENDIX

## MODEL-BASED CAUSAL BAYESIAN OPTIMIZATION

### A.1 BACKGROUND ON GPS

Here we give more background on the vector-valued GP models we use. We will drop the $i$-node index and consider the modelling of a single function with scalar output $f : \mathcal{A} \times \mathcal{Z} \to \mathbb{R}$ in RKHS $\mathcal{H}_k$ with kernel $k : \mathcal{S} \times \mathcal{S} \to \mathbb{R}$ where $\mathcal{S} = (\mathcal{Z} \times \mathcal{A})$. We assume measurement noise with variance $\rho^2$. Assuming prior mean and variance $\mu_0, \sigma_0$ and dataset $\mathcal{D}_t = \{ \boldsymbol{z}_{1:t}, \boldsymbol{a}_{1:t}, \boldsymbol{x}_{1:t} \}$ where $\boldsymbol{x}_t \in \mathbb{R}$, we get a posterior GP mean and variance given by updates

$$\mu_t(\boldsymbol{z}, \boldsymbol{a}) = \boldsymbol{k}_t(\boldsymbol{z}, \boldsymbol{a})^\top \left( \boldsymbol{K}_t + \rho^2 \boldsymbol{I} \right)^{-1} \boldsymbol{x}_{1:t} \,, \tag{13}$$

$$\sigma_t^2(\boldsymbol{z}, \boldsymbol{a}) = k((\boldsymbol{z}, \boldsymbol{a}); (\boldsymbol{z}, \boldsymbol{a})) - \boldsymbol{k}_t(\boldsymbol{z}, \boldsymbol{a})^\top \left( \boldsymbol{K}_t + \rho^2 \boldsymbol{I} \right)^{-1} \boldsymbol{k}_t(\boldsymbol{z}, \boldsymbol{a}) \,, \tag{14}$$

where

$$(\boldsymbol{K}_t)_{t_1, t_2} = k((\boldsymbol{z}_{t_1}, \boldsymbol{a}_{t_1}); (\boldsymbol{z}_{t_2}, \boldsymbol{a}_{t_2})),$$
$$\boldsymbol{k}_t(\boldsymbol{z}, \boldsymbol{a}) = [k((\boldsymbol{z}_1, \boldsymbol{a}_1); (\boldsymbol{z}, \boldsymbol{a})) \ldots, k((\boldsymbol{z}_t, \boldsymbol{a}_t); (\boldsymbol{z}, \boldsymbol{a}))] \,,$$

and $\boldsymbol{I}$ is the identity matrix. This follows the standard GP update given in Rasmussen (2003).

In this work, we model functions with vector-valued outputs $\boldsymbol{f} : \mathcal{A} \times \mathcal{Z} \to \mathcal{X} \in \mathbb{R}^d$ (we will continue to drop the $i$ index). For this we follow Chowdhury & Gopalan (2019) and use a scalar-output GP that takes the component of the output vector under consideration as an input. That is, we model $f(\cdot, l)$ (the $l$th output component of the function under study) where $\boldsymbol{f} = [f(\cdot, 1), \ldots, f(\cdot, d)]$. We again assume that $f$ is from RKHS $\mathcal{H}_k$ but with kernel $k : \tilde{\mathcal{S}} \times \tilde{\mathcal{S}} \to \mathbb{R}$ where $\tilde{\mathcal{S}} = (\mathcal{Z} \times \mathcal{A} \times [d])$. Assuming prior mean and variance $\mu_0, \sigma_0$ and dataset $\mathcal{D}_t = \{ \boldsymbol{z}_{1:t}, \boldsymbol{a}_{1:t}, \boldsymbol{x}_{1:t} \}$ where $\boldsymbol{x}_t \in \mathbb{R}^d$, we get a posterior GP mean and variance given by updates

$$\mu_t(\boldsymbol{z}, \boldsymbol{a}, l) = \boldsymbol{k}_t(\boldsymbol{z}, \boldsymbol{a}, l)^\top (\boldsymbol{K}_t + \rho^2 \boldsymbol{I})^{-1} \mathrm{vec}(\boldsymbol{x}_{1:t}) \,, \tag{15}$$

$$\sigma_t^2(\boldsymbol{z}, \boldsymbol{a}, l) = k((\boldsymbol{z}, \boldsymbol{a}, l); (\boldsymbol{z}, \boldsymbol{a}, l)) - \boldsymbol{k}_t(\boldsymbol{z}, \boldsymbol{a}, l)^\top (\boldsymbol{K}_t + \rho^2 \boldsymbol{I})^{-1} \boldsymbol{k}_t(\boldsymbol{z}, \boldsymbol{a}, l) \,, \tag{16}$$

where $\mathrm{vec}(\boldsymbol{x}_{1:t}) = [\boldsymbol{x}_{1,1}, \boldsymbol{x}_{1,2}, \ldots, \boldsymbol{x}_{t,d}]^\top$ and for $(t_1, l), (t_2, l') \in [(1,1), (1,2), \ldots, (t, d)]$:

$$[\boldsymbol{K}_t]_{(t_1, l), (t_2, l')} = k((\boldsymbol{z}_{t_1, l}, \boldsymbol{a}_{t_1, l}, l); (\boldsymbol{z}_{t_2, l'}, \boldsymbol{a}_{t_2, l'}, l')),$$
$$\boldsymbol{k}_t(\boldsymbol{z}, \boldsymbol{a}, l)^\top = [k((\boldsymbol{z}_{1,1}, \boldsymbol{a}_{1,1}, 1); (\boldsymbol{z}, \boldsymbol{a}, l)), \ldots, k((\boldsymbol{z}_{t,d}, \boldsymbol{a}_{t,d}, d); (\boldsymbol{z}, \boldsymbol{a}, l))]^\top \,.$$

This follows the posterior update of Chowdhury & Gopalan (2019). The key idea is to use a single scalar-output GP with kernel $k$ for modeling all output components, but introduce the component index as part of the input space. This requires introducing the notation $\mathrm{vec}(\boldsymbol{x}_{1:t})$ and ordering all observations, of all time points and all components, in a single index, since the GP update for component $l$ considers observations of all other components. Under this GP model the components of $\boldsymbol{f}$ need not be independent if kernel $k$ is designed to model dependency between the components. In our work, we use one of these vector-valued GP models for each individual random variable in our causal model, leading to the reintroduction of the additional $i$ index in Eqs. (7) and (8).

### A.2 PROOFS FOR THE THEORETICAL ANALYSIS

Our analysis closely follows Curi et al. (2020), particularly the proofs in their Appendix D, where they prove similar guarantees for a model-based reinforcement learning problem. In contrast to Curi et al. (2020), which models RL transition dynamics with a single GP for all timesteps, MCBO uses independent GPs for modeling the functional relation $\{f_1, \ldots, f_m\}$ and uses the causal graph $\mathcal{G}$ to determine the input and output variables of these functions.

This section is organized as follows. In Appendix A.2.1, we discuss a notion of model complexity $\Gamma_T$ similar to the one introduced in the RL setting by Curi et al. (2020). We then bound the cumulative regret in terms of $\Gamma_T$ in Appendix A.2.2. Finally, in Appendix A.2.3, we prove Theorem 1 by connecting our notion of model complexity with the maximum information gain of a GP model.

The norm notation $\|\cdot\|$ refers to $\ell_2$-norm if no additional notation is given. We let $\{\boldsymbol{a}_{i,t} \in \mathcal{A}_i, \boldsymbol{z}_{i,t} \in \mathcal{Z}_i\}_{i, t > 0}$ denote the set of actions chosen by MCBO and the realizations of the parents of node $i$, respectively.

A.2.1 MODEL COMPLEXITY

The number of samples needed to learn a low-regret action is related to the number of samples needed to learn all GP models in our SCM. This is analogous to the classic BO setting (Srinivas et al., 2010). We quantify the model complexity of our entire model class $\mathcal{M}_T$ as

$$\Gamma_T = \max_{(\boldsymbol{z},\boldsymbol{a}) \in A \subset \{\mathcal{Z} \times \mathcal{A}\}^T} \sum_{t=1}^{T} \sum_{i=0}^{m} \left\| \boldsymbol{\sigma}_{i,t-1}(\boldsymbol{z}_{i,t}, \boldsymbol{a}_{i,t}) \right\|^2. \tag{17}$$

This measure of model complexity closely relates to the *maximum information gain* $\gamma_T$ used for proving regret guarantees in BO (Srinivas et al., 2010). For a single GP model, $\gamma_T$ is the maximum information gain about the unknown $f$ that can be obtained from noisy evaluations of the $f$ at fixed inputs (see Eq. (39)). Later we show that $\Gamma_T$ can be bounded by a sum of the information gains for all $m$ GPs. It is worth noting that Equation 17 may be a loose notion of model complexity because it assumes we can independently choose every $\boldsymbol{z}_i$, but for many graphs, there could be overlap in the $\boldsymbol{z}_i, \boldsymbol{z}_j$ for $i \neq j$ (two nodes could have a shared parent).

A.2.2 ANALYSIS IN TERMS OF GENERAL MODEL COMPLEXITY $\Gamma_T$

In this section, we will prove a theorem similar to Theorem 1 but in terms of the model complexity we define in Eq. (17). Note that this version of the theorem does not require that $\boldsymbol{\mu}$ and $\boldsymbol{\sigma}$ come from a GP model with independent outputs, but any model such that Assumptions 1–3 are satisfied. In later sections, when using a GP model we bound $\Gamma_T$ in terms of the maximum information gain of the $m$ GPs used to get Theorem 1.

We will use the function $\boldsymbol{\Sigma}_{i,t}(\cdot)$ to represent a matrix of all zeros except the values of the diagonal, given by $\boldsymbol{\sigma}_{i,t}(\cdot)$. A a result, $\boldsymbol{\sigma}_{i,t}(\cdot) = \text{diag}\left(\boldsymbol{\Sigma}_{i,t}(\cdot)\right)$

**Theorem 2.** *Consider the optimization problem in Eq. (4) with SCM satisfying Assumptions 1–3 where $\mathcal{G}$ is known but $\boldsymbol{F}$ is unknown. Then, for all $T \geq 1$, with probability at least $1 - \delta$, the regret of Algorithm 1 is bounded by*

$$R_T \leq \mathcal{O}\left( L_f^N L_\sigma^N \beta_T^N K^N \sqrt{Tm\,\Gamma_T} \right).$$

We first sketch the proof steps. In Lemma 1 we show that with, high probability, there exists some set of functions $\boldsymbol{\eta}$ that allows the reparameterized plausible SCM model in Eq. (11) to match the true SCM. Recall the mechanism of the ground-truth SCM in Eq. (2)

$$\boldsymbol{x}_{i,t} = \boldsymbol{f}_i(\boldsymbol{z}_{i,t}, \boldsymbol{a}_{i,t}) + \boldsymbol{\omega}_{i,t}, \ \ \forall i \in \{0, \dots, m\}, \tag{2}$$

and the mechanism of the optimistic SCM model using the reparameterization of Eq. (11)

$$\tilde{\boldsymbol{x}}_{i,t} = \boldsymbol{f}_i(\tilde{\boldsymbol{z}}_{i,t}, \tilde{\boldsymbol{a}}_{i,t}) + \tilde{\boldsymbol{\omega}}_{i,t} \tag{18}$$
$$= \boldsymbol{\mu}_{i,t-1}(\tilde{\boldsymbol{z}}_{i,t}, \tilde{\boldsymbol{a}}_{i,t}) + \beta_t \boldsymbol{\Sigma}_{i,t-1}(\tilde{\boldsymbol{z}}_{i,t}, \tilde{\boldsymbol{a}}_{i,t})\boldsymbol{\eta}_i(\tilde{\boldsymbol{z}}_{i,t}, \tilde{\boldsymbol{a}}_{i,t}) + \tilde{\boldsymbol{\omega}}_{i,t}, \ \ \forall i \in \{0, \dots, m\}. \tag{19}$$

In Lemma 2, Lemma 3 and Corollary 1 we bound the instantaneous regret (regret at some specific timepoint $t$) by bounding the difference in SCM output, for the same action input, assuming the true SCM vs the optimistic reparameterized SCM. Then in Lemma 5 we use the intermediate result of Lemma 4 to show that our bound on instantaneous regret implies a bound on cumulative regret.

In Lemma 1, for convenience, we will drop the explicit dependence of all quantities on $t$.

**Lemma 1.** *Assume some fixed set of actions $\boldsymbol{a}$ is chosen at any timepoint $t$. Under Assumption 3, for any $\boldsymbol{x}$ generated by the true SCM Eq. (2), with probability at least $1 - \delta$ there exists a set of functions $\boldsymbol{\eta} = \{\eta_i\}_{i=0}^{m}$, where $\eta_i \colon \mathcal{Z}_i \times \mathcal{A}_i \to [-1, 1]^d$, such that $\boldsymbol{x} = \tilde{\boldsymbol{x}}$ if $\forall i \ \ \boldsymbol{\omega}_i = \tilde{\boldsymbol{\omega}}_i$.*

*Proof.* Since $\boldsymbol{\omega} = \tilde{\boldsymbol{\omega}}$ we only need to prove that there exists some $\boldsymbol{\eta}$ such that $\forall i \ \ \boldsymbol{f}_i(\boldsymbol{z}_i, \boldsymbol{a}_i) = \boldsymbol{\mu}_i(\tilde{\boldsymbol{z}}_i, \tilde{\boldsymbol{a}}_i) + \beta \boldsymbol{\Sigma}_i(\tilde{\boldsymbol{z}}_i, \tilde{\boldsymbol{a}}_i)\boldsymbol{\eta_i}$.

By Assumption 3, with probability $1 - \delta$, for all $i = 0, \ldots, m$, we have an elementwise bound $|\boldsymbol{f}_i(\boldsymbol{z}_i, \boldsymbol{a}_i) - \boldsymbol{\mu}_i(\boldsymbol{z}_i, \boldsymbol{a}_i)| \leq \beta_i \boldsymbol{\sigma_i}(\boldsymbol{z}_{i,t}, \boldsymbol{a}_{i,t})$. Thus for each $\boldsymbol{z}_{i,t}, \boldsymbol{a}_{i,t}$ there exists a vector $\boldsymbol{\eta}_i$ with values in $[-1, 1]^d$ such that $\boldsymbol{f}_i(\boldsymbol{z}_i, \boldsymbol{a}_i) = \boldsymbol{\mu}_i(\boldsymbol{z}_i, \boldsymbol{a}_i) + \beta \boldsymbol{\Sigma}_i(\boldsymbol{z}_i, \boldsymbol{a}_i) \boldsymbol{\eta}_i$. Note that this is not quite what we need because the RHS contains $\boldsymbol{z}_i$ and not $\tilde{\boldsymbol{z}}_i$. We will now use an inductive argument on $i$ that constructs each $\boldsymbol{\eta}_i$ sequentially from $i = 0$ to $m$.

Base case: we must prove that for $i = 0$ we have $\boldsymbol{f}_0(\boldsymbol{z}_0, \boldsymbol{a}_0) = \boldsymbol{\mu}_0(\tilde{\boldsymbol{z}}_0, \tilde{\boldsymbol{a}}_0) + \beta_t \boldsymbol{\Sigma}_0(\tilde{\boldsymbol{z}}_0, \tilde{\boldsymbol{a}}_0) \boldsymbol{\eta_0}$. We know $\boldsymbol{z}_0 = \tilde{\boldsymbol{z}}_0$ (both are the empty vector) and $\boldsymbol{a} = \tilde{\boldsymbol{a}}$ by the assumption of a fixed action. Then there exists some vector $\boldsymbol{\eta}_0$ such that $\boldsymbol{f}_0(\boldsymbol{z}_0, \boldsymbol{a}_0) = \boldsymbol{\mu}_0(\boldsymbol{z}_0, \boldsymbol{a}_0) + \beta \boldsymbol{\Sigma}_0(\boldsymbol{z}_0, \boldsymbol{a}_0) \boldsymbol{\eta_0} = \boldsymbol{\mu}_0(\tilde{\boldsymbol{z}}_0, \tilde{\boldsymbol{a}}_0) + \beta \boldsymbol{\Sigma}_0(\tilde{\boldsymbol{z}}_0, \tilde{\boldsymbol{a}}_0) \boldsymbol{\eta_0}$. Let $\boldsymbol{\eta}_0(\cdot)$ be the function that outputs the vector $\boldsymbol{\eta}_0$ given input $\boldsymbol{z}_0, \boldsymbol{a}_0$ and the base case is proven.

Now assume the inductive hypothesis: $\forall j < i$ we have $\boldsymbol{f}_j(\boldsymbol{z}_j, \boldsymbol{a}_j) = \boldsymbol{\mu}_j(\tilde{\boldsymbol{z}}_j, \tilde{\boldsymbol{a}}_j) + \beta \boldsymbol{\Sigma}_j(\tilde{\boldsymbol{z}}_j, \tilde{\boldsymbol{a}}_j) \boldsymbol{\eta_j}$. We want to show that this implies $\boldsymbol{f}_i(\boldsymbol{z}_i, \boldsymbol{a}_i) = \boldsymbol{\mu}_i(\tilde{\boldsymbol{z}}_i, \tilde{\boldsymbol{a}}_i) + \beta \boldsymbol{\Sigma}_i(\tilde{\boldsymbol{z}}_i, \tilde{\boldsymbol{a}}_i) \boldsymbol{\eta}_i$. We know $\boldsymbol{a}_i = \tilde{\boldsymbol{a}}_i$ by the assumption of a fixed action. $\boldsymbol{z}_i = \tilde{\boldsymbol{z}}_i$ because $\tilde{\boldsymbol{z}}_i = [\tilde{\boldsymbol{x}}_{pa_i[1]}, \ldots, \tilde{\boldsymbol{x}}_{pa_i[|pa_i|]}]^T$ and we selected each $\boldsymbol{\eta}_j$ such that $\tilde{\boldsymbol{x}}_j = \boldsymbol{x}_j$. Then there exists some vector $\boldsymbol{\eta}_i$ such that $\boldsymbol{f}_i(\boldsymbol{z}_i, \boldsymbol{a}_i) = \boldsymbol{\mu}_i(\boldsymbol{z}_i, \boldsymbol{a}_i) + \beta \boldsymbol{\Sigma}_i(\boldsymbol{z}_i, \boldsymbol{a}_i)) \boldsymbol{\eta_i} = \boldsymbol{\mu}_i(\tilde{\boldsymbol{z}}_i, \tilde{\boldsymbol{a}}_i) + \beta \boldsymbol{\Sigma}_i(\tilde{\boldsymbol{z}}_i, \tilde{\boldsymbol{a}}_i)) \boldsymbol{\eta_i}$. Let $\boldsymbol{\eta}_i(\cdot)$ output the vector $\boldsymbol{\eta}_i$ given input $\boldsymbol{z}_i, \boldsymbol{a}_i$ and the inductive step is proven. $\qquad\square$

**Lemma 2.** *Under Assumption 3, with probability at least $(1 - \delta)$ $\forall t \geq 0$ the instantaneous regret $r_t$ is bounded by*

$$r_t = \mathbb{E}[y|\boldsymbol{F}, \boldsymbol{a}^*] - \mathbb{E}[y|\boldsymbol{F}, \boldsymbol{a}_{:,t}] \leq \mathbb{E}\left[y|\tilde{\boldsymbol{F}}_t, \boldsymbol{a}_{:,t}\right] - \mathbb{E}[y|\boldsymbol{F}, \boldsymbol{a}_{:,t}]. \tag{20}$$

*Proof.* The result follows directly from,

$$\mathbb{E}[y|\boldsymbol{F}, \boldsymbol{a}^*] \leq \mathbb{E}\left[y|\tilde{\boldsymbol{F}}_t, \boldsymbol{a}_{:,t}\right]. \tag{21}$$

This is true by the definition of $\tilde{\boldsymbol{F}}_t, \boldsymbol{a}_{:,t}$ as the argmax of Eq. (12) and that with probability at least $(1 - \delta)$ we have $\boldsymbol{F} \in \mathcal{M}_T$. $\qquad\square$

We now show how the observations under the true and optimistic dynamics differ for a fixed noise sequence $\tilde{\boldsymbol{\omega}} = \boldsymbol{\omega}$ and the fixed action $\boldsymbol{a}_{i,t}$ at any time $t$.

**Lemma 3.** *Under Assumptions 1–3, let $\bar{L}_{f,t} = 1 + L_{\mathrm{f}} + 2\beta_t L_\sigma$. Then, for all iterations $t > 0$, any functions $\boldsymbol{\eta}_i \colon \mathbb{R}^{p_i} \times \mathbb{R}^{q_i} \to [-1, 1]^{d_i}$ and any sequence of $\boldsymbol{\omega}_i$ with $\tilde{\boldsymbol{\omega}}_i = \boldsymbol{\omega}_i$ (for all $i$), we have*

$$\|\boldsymbol{x}_{m,t} - \tilde{\boldsymbol{x}}_{m,t}\| \leq 2\beta_t K^N \bar{L}_{f,t}^N \sum_{i=0}^m \|\boldsymbol{\sigma}_{i,t-1}(\boldsymbol{z}_{i,t}, \boldsymbol{a}_{i,t})\| \tag{22}$$

*Proof.* We prove by induction on $i$.

*Base case.* Consider the base case $i = 0$. Because the nodes are topologically ordered we will have $pa_0 = \emptyset$. Its realization, therefore, depends only on the chosen action. Formally, we assume $\boldsymbol{z}_0 = \emptyset$, $\boldsymbol{x}_0 = \tilde{\boldsymbol{f}}_0(\boldsymbol{z}_0, \boldsymbol{a}_0) + \boldsymbol{\omega}_0$ and $\tilde{\boldsymbol{x}}_0 = \tilde{\boldsymbol{f}}_0(\tilde{\boldsymbol{z}}_0, \tilde{\boldsymbol{a}}_0) + \tilde{\boldsymbol{\omega}}_0$. Since $\boldsymbol{\omega}_0 = \tilde{\boldsymbol{\omega}}_0$,

$$\begin{aligned}
\|\boldsymbol{x}_{0,t} - \tilde{\boldsymbol{x}}_{0,t}\| &= \|\boldsymbol{f}_0(\boldsymbol{z}_{0,t}, \boldsymbol{a}_{0,t}) + \boldsymbol{\omega}_{0,t} - \boldsymbol{\mu}_{0,t-1}(\boldsymbol{z}_{0,t}, \boldsymbol{a}_{0,t}) - \beta_t \boldsymbol{\Sigma}_{0,t-1}(\boldsymbol{z}_{0,t}, \boldsymbol{a}_{0,t}) \boldsymbol{\eta}_0(\boldsymbol{z}_{0,t}, \boldsymbol{a}_{0,t}) - \tilde{\boldsymbol{\omega}}_0\| \\
&\leq \|\boldsymbol{f}_0(\boldsymbol{z}_{0,t}, \boldsymbol{a}_{0,t}) - \boldsymbol{\mu}_{0,t-1}(\boldsymbol{z}_{0,t}, \boldsymbol{a}_{0,t})\| + \|\beta_t \boldsymbol{\Sigma}_{0,t-1}(\boldsymbol{z}_{0,t}, \boldsymbol{a}_0) \boldsymbol{\eta}_0(\boldsymbol{z}_{0,t}, \boldsymbol{a}_{0,t})\| \\
&\leq 2\beta_t \|\boldsymbol{\Sigma}_{0,t-1}(\boldsymbol{z}_{0,t}, \boldsymbol{a}_{0,t})\|
\end{aligned}$$

In the following, we omit the dependence on the action $\boldsymbol{a}$, e.g., using $\boldsymbol{f}_i(\boldsymbol{z}_{i,t})$ instead of $\boldsymbol{f}_i(\boldsymbol{z}_{i,t}, \boldsymbol{a}_{i,t})$ since we assume the actions to be the same for the process generating $\boldsymbol{x}_{i,t}$ and $\tilde{\boldsymbol{x}}_{i,t}$.

*Induction step.* Now assuming that $\|\boldsymbol{x}_{i-1,t} - \tilde{\boldsymbol{x}}_{i-1,t}\| \leq 2\beta_t K^{N_{i-1}} \bar{L}_{f,t}^{N_{i-1}} \sum_{j=0}^{i-1} \|\boldsymbol{\sigma}_{j,t-1}(\boldsymbol{z}_{j,t})\|$ we prove a similar result for the $i$th node.

$$\|\boldsymbol{x}_{i,t} - \tilde{\boldsymbol{x}}_{i,t}\| \overset{\textcircled{1}}{=} \|\boldsymbol{f}_i(\boldsymbol{z}_{i,t}) + \boldsymbol{\omega}_{i,t} - \boldsymbol{\mu}_{i,t-1}(\tilde{\boldsymbol{z}}_{i,t}) - \beta_t \boldsymbol{\Sigma}_{i,t-1}(\tilde{\boldsymbol{z}}_{i,t}) \boldsymbol{\eta}_i(\tilde{\boldsymbol{z}}_{i,t}) - \tilde{\boldsymbol{\omega}}_{i,t}\|$$

$$\overset{(2)}{=} \|\boldsymbol{f}_i(\boldsymbol{z}_{i,t}) - \boldsymbol{\mu}_{i,t-1}(\tilde{\boldsymbol{z}}_{i,t}) - \beta_t \boldsymbol{\Sigma}_{i,t-1}(\tilde{\boldsymbol{z}}_{i,t})\boldsymbol{\eta}_i(\tilde{\boldsymbol{z}}_{i,t}) + \boldsymbol{f}_i(\tilde{\boldsymbol{z}}_{i,t}) - \boldsymbol{f}_i(\tilde{\boldsymbol{z}}_{i,t})\|$$

$$\overset{(3)}{=} \|\boldsymbol{f}_i(\tilde{\boldsymbol{z}}_{i,t}) - \boldsymbol{\mu}_{i,t-1}(\tilde{\boldsymbol{z}}_{i,t}) - \beta_t \boldsymbol{\Sigma}_{i,t-1}(\tilde{\boldsymbol{z}}_{i,t})\boldsymbol{\eta}_i(\tilde{\boldsymbol{z}}_{i,t}) + \boldsymbol{f}_i(\boldsymbol{z}_{i,t}) - \boldsymbol{f}_i(\tilde{\boldsymbol{z}}_{i,t})\|$$

$$\overset{(4)}{\leq} \|\boldsymbol{f}_i(\tilde{\boldsymbol{z}}_{i,t}) - \boldsymbol{\mu}_{i,t-1}(\tilde{\boldsymbol{z}}_{i,t})\| + \|\beta_t \boldsymbol{\Sigma}_{i,t-1}(\tilde{\boldsymbol{z}}_{i,t})\boldsymbol{\eta}_i(\tilde{\boldsymbol{z}}_{i,t})\| + \|\boldsymbol{f}_i(\boldsymbol{z}_{i,t}) - \boldsymbol{f}_i(\tilde{\boldsymbol{z}}_{i,t})\|$$

$$\overset{(5)}{\leq} \beta_t \|\boldsymbol{\sigma}_{i,t-1}(\tilde{\boldsymbol{z}}_{i,t})\| + \beta_t \|\boldsymbol{\sigma}_{i,t-1}(\tilde{\boldsymbol{z}}_{i,t})\| + L_f \|\boldsymbol{z}_{i,t} - \tilde{\boldsymbol{z}}_{i,t}\|$$

$$\overset{(6)}{=} 2\beta_t \|\boldsymbol{\sigma}_{i,t-1}(\tilde{\boldsymbol{z}}_{i,t})\| + L_f \|\boldsymbol{z}_{i,t} - \tilde{\boldsymbol{z}}_{i,t}\|$$

$$\overset{(7)}{=} 2\beta_t \|\boldsymbol{\sigma}_{i,t-1}(\tilde{\boldsymbol{z}}_{i,t}) + \boldsymbol{\sigma}_{i,t-1}(\boldsymbol{z}_{i,t}) - \boldsymbol{\sigma}_{i,t-1}(\boldsymbol{z}_{i,t})\| + L_f \|\boldsymbol{z}_{i,t} - \tilde{\boldsymbol{z}}_{i,t}\|$$

$$\overset{(8)}{\leq} 2\beta_t \left( \|\boldsymbol{\sigma}_{i,t-1}(\boldsymbol{z}_{i,t})\| + L_\sigma \|\boldsymbol{z}_{i,t} - \tilde{\boldsymbol{z}}_{i,t}\| \right) + L_f \|\boldsymbol{z}_{i,t} - \tilde{\boldsymbol{z}}_{i,t}\|$$

$$\overset{(9)}{\leq} 2\beta_t \|\boldsymbol{\sigma}_{i,t-1}(\boldsymbol{z}_{i,t})\| + \left(1 + L_f + 2\beta_t L_\sigma\right) \|\boldsymbol{z}_{i,t} - \tilde{\boldsymbol{z}}_{i,t}\|$$

$$\overset{(10)}{\leq} 2\beta_t \|\boldsymbol{\sigma}_{i,t-1}(\boldsymbol{z}_{i,t})\| + \left(1 + L_f + 2\beta_t L_\sigma\right) \sum_{j \in pa_i} \|\boldsymbol{x}_{i,t} - \tilde{\boldsymbol{x}}_{i,t}\|$$

$$\overset{(11)}{\leq} 2\beta_t \|\boldsymbol{\sigma}_{i,t-1}(\boldsymbol{z}_{i,t})\|$$
$$+ \left(1 + L_f + 2\beta_t L_\sigma\right) \sum_{j \in pa_i} 2\beta_t K^{N_j} \underbrace{\left(1 + L_f + 2\beta_t L_\sigma\right)^{N_j}}_{=: \bar{L}_f} \sum_{h=0}^{j} \|\boldsymbol{\sigma}_{h,t-1}(\boldsymbol{z}_{h,t})\|$$

$$\overset{(12)}{\leq} 2\beta_t K^{N_i} \bar{L}_{f,t}^{N_i} \sum_{j=0}^{i} \|\boldsymbol{\sigma}_{j,t-1}(\boldsymbol{z}_{j,t})\| \tag{23}$$

where ① follows the dynamics Eqs. (2) and (11). In ②, we assume the noise to be equal and add and subtract the same term. In ③ and ④, we reorder terms and apply the triangle inequality. In ⑤ and ⑥, we rely on the calibrated uncertainty and Lipschitz dynamics, then collect terms and use diagonality of the matrix $\boldsymbol{\Sigma}_{i,t-1}(\cdot)$. In ⑦ and ⑧, we add and subtract the same term and use the Lipschitz continuity of $\boldsymbol{\sigma}_{i,t-1}$. Finally, in ⑨, we add 1 to ensure that we can later upper bound this term by taking the exponential of it. ⑩ applies the triangle inequality. ⑪ follows the inductive hypothesis, and ⑫ is due to the depth of at least one parent $j$ being $N_j = N_i - 1$. $\qquad\square$

Now we will relate this bound on the observations to a bound on $y_t$ when selecting actions according to MCBO in both the optimistic and true dynamics.

**Corollary 1.** *Under the assumptions of Lemma 3, for any sequence of $\eta_i \in [-1,1]^{d_i}$, $\boldsymbol{\theta} \in \mathcal{D}$, and $t \geq 1$ we have that*

$$\mathbb{E}\left[y_t | \tilde{\boldsymbol{F}}_t, \boldsymbol{a}_{:,t}\right] - \mathbb{E}[y_t | \boldsymbol{F}, \boldsymbol{a}_{:,t}] \leq 2\beta_t K^N \bar{L}_{f,t}^N \mathbb{E}_{\omega = \tilde{\omega}}\left[ \sum_{i=0}^{m} \|\boldsymbol{\sigma}_{i,t-1}(\boldsymbol{z}_{i,t}, \boldsymbol{a}_{i,t})\| \right] \tag{24}$$

*Proof.* This follows from Lemma 3. $Y$ is just the final observation so

$$\mathbb{E}\left[y_t | \tilde{\boldsymbol{F}}_t, \boldsymbol{a}_{:,t}\right] - \mathbb{E}[y_t | \boldsymbol{F}, \boldsymbol{a}_{:,t}] = \mathbb{E}[\|\boldsymbol{x}_{m,t} - \tilde{\boldsymbol{x}}_{m,t}\| \mid \boldsymbol{a}_{:,t}]$$

$$\leq 2\beta_t K^N \bar{L}_{f,t}^N \mathbb{E}\left[ \sum_{i=0}^{m} \|\boldsymbol{\sigma}_{i,t-1}(\boldsymbol{z}_{i,t}, \boldsymbol{a}_{i,t})\| \right]$$

$\qquad\square$

**Lemma 4.** *Under Assumption 3, let $L_{Y,t} = 2\beta_t \bar{L}_{f,t}^N$. Then, with probability at least $(1-\delta)$ it holds for all $t \geq 0$ that*

$$r_t^2 \leq L_{Y,t}^2 K^{2N} m \mathbb{E}\left[\sum_{i=0}^m \|\boldsymbol{\sigma}_{i,t-1}(\boldsymbol{z}_{i,t}, \boldsymbol{a}_{i,t})\|_2^2\right] \tag{25}$$

*Proof.*

$$r_t \leq \mathbb{E}\left[y_t | \tilde{\boldsymbol{F}}_t, \boldsymbol{a}_t\right] - \mathbb{E}[y_t | \boldsymbol{F}, \boldsymbol{a}_t] \tag{26}$$

$$\leq 2\beta_t K^N \bar{L}_{f,t}^N \mathbb{E}\left[\sum_{i=0}^m \|\boldsymbol{\sigma}_{i,t-1}(\boldsymbol{z}_{i,t}, \boldsymbol{a}_{i,t})\|\right] \tag{27}$$

$$\leq L_{Y,t} K^N \mathbb{E}\left[\sum_{i=0}^m \|\boldsymbol{\sigma}_{i,t-1}(\boldsymbol{z}_{i,t}, \boldsymbol{a}_{i,t})\|\right] \tag{28}$$

$$r_t^2 \leq L_{Y,t}^2 K^{2N} \left(\mathbb{E}\left[\sum_{i=0}^m \|\boldsymbol{\sigma}_{i,t-1}(\boldsymbol{z}_{i,t}, \boldsymbol{a}_{i,t})\|\right]\right)^2 \tag{29}$$

$$\leq L_{Y,t}^2 K^{2N} \mathbb{E}\left[\left(\sum_{i=0}^m \|\boldsymbol{\sigma}_{i,t-1}(\boldsymbol{z}_{i,t}, \boldsymbol{a}_{i,t})\|\right)^2\right] \tag{30}$$

$$\leq L_{Y,t}^2 K^{2N} m \mathbb{E}\left[\sum_{i=0}^m \|\boldsymbol{\sigma}_{i,t-1}(\boldsymbol{z}_{i,t}, \boldsymbol{a}_{i,t})\|_2^2\right] \tag{31}$$

The last two lines are Jensen's inequality. $\qquad\square$

Now we bound cumulative regret $R_T$.

**Lemma 5.** *Under the assumption of Assumptions 1–3, with probability at least $(1-\delta)$ it holds for all $t \geq 0$ that*

$$R_T^2 \leq T L_{Y,T}^2 m \sum_{t=1}^T \mathbb{E}\left[\sum_{i=0}^m \|\boldsymbol{\sigma}_{i,t}(\boldsymbol{z}_{i,t}, \boldsymbol{a}_{i,t})^2\|_2^2\right] \tag{32}$$

*Proof.*

$$R_T^2 = \left(\sum_{t=1}^T r_t\right)^2 \overset{\textcircled{1}}{\leq} T \sum_{t=1}^T r_t^2 \overset{\textcircled{2}}{\leq} T L_{Y,T}^2 m \sum_{t=1}^T \mathbb{E}\left[\sum_{i=0}^m \|\boldsymbol{\sigma}_{i,t}(\boldsymbol{z}_{i,t}, \boldsymbol{a}_{i,t})^2\|_2^2\right], \tag{33}$$

$$\square$$

where $\textcircled{1}$ is due to Jensen's inequality and $\textcircled{2}$ follows Lemma 4. Similar to the equivalent lemma in Curi et al. (2020), this bound is dependent on the data observed by the iteration $t$, making it hard to interpret in a general case. To this end, we further provide the worst-case bound dependent on the model complexity $\Gamma_T$.

**Lemma 6.** *Under Assumptions 1–3, with probability at least $(1-\delta)$ it holds for all $t \geq 0$ that*

$$R_T^2 \leq T L_{Y,T}^2 K^{2N} m \Gamma_T \tag{34}$$

*Proof.* Substituting in Eq. (17) we have

$$\sum_{t=1}^T \mathbb{E}\left[\sum_{i=0}^m \left\|\sigma_{i,t}(\boldsymbol{z}_{i,t}, \boldsymbol{a}_{i,t})^2\right\|^2\right] \leq \Gamma_T \tag{35}$$

and the result follows. $\qquad\square$

Taking square roots and substituting in for $L_{Y,T}$ in terms of $\beta_T$, $L_f$ and $L_\sigma$ in Eq. (34) concludes the proof for Theorem 2.

A.2.3 PROOF OF THEOREM 1

We can bound $\Gamma_T$ in Theorem 2 to get a bound that depends on the specific GP model used for each $\boldsymbol{f}_i$. We can show that for many commonly used kernels MCBO achieves sublinear (in $T$) regret.

**Theorem 1.** *Consider the optimization problem in Eq. (4) with SCM satisfying Assumptions 1–3 where $\mathcal{G}$ is known but $\boldsymbol{F}$ is unknown. Then for all $T \geq 1$, with probability at least $1 - \delta$, the cumulative regret of Algorithm 1 is bounded by*

$$R_T \leq \mathcal{O}\left( L_f^N L_\sigma^N \beta_T^N K^N m \sqrt{T \gamma_T} \right).$$

*Proof.* **Step 1.** *Some preliminaries relating mutual information $I_T(\boldsymbol{x}_{i,1:T}, f_{i,1:T})$ and maximum information gain $\gamma_T$*

In the following, we consider the information gain for the node $i$, i.e., for $\boldsymbol{x}_{i,1:T} \in \mathbb{R}^{d \times T}$ and $\boldsymbol{f}_{i,1:T} = [\boldsymbol{f}_i(\boldsymbol{z}_{i,1}, \boldsymbol{a}_{i,1}), \ldots, \boldsymbol{f}_i(\boldsymbol{z}_{i,T}, \boldsymbol{a}_{i,T})]^\top$ evaluated at points $A_i = \{\boldsymbol{z}_{i,1:T}, \boldsymbol{a}_{i,1:T}\}$, $A_i \in \mathcal{Z}_i \times \mathcal{A}_i$. In this step, for simplicity, when clear from context we will omit the $i$-notation in the derivation for the information gain. Mutual information $I_T(\boldsymbol{x}_{1:T}, \boldsymbol{f}_{1:T})$ is then defined as:

$$I(\boldsymbol{x}_{1:T}, \boldsymbol{f}_{1:T}) = H(\boldsymbol{x}_{1:T}) - H(\boldsymbol{x}_{1:T} | \boldsymbol{f}_{1:T}),$$

where $H(\cdot)$ denotes entropy. For fitting the GPs, the following models are used: $\boldsymbol{x}_t | \boldsymbol{f}_t \sim \mathcal{N}(\boldsymbol{f}_t(\boldsymbol{z}_t, \boldsymbol{a}_t), \rho^2 \boldsymbol{I})$ and $\boldsymbol{x}_t | \boldsymbol{x}_{1:t-1}, \boldsymbol{z}_{1:t}, \boldsymbol{a}_{1:t} \sim \mathcal{N}(\boldsymbol{\mu}_{t-1}(\boldsymbol{z}_t, \boldsymbol{a}_t), \rho \boldsymbol{I} + \boldsymbol{\Sigma}_{t-1}(\boldsymbol{z}_t, \boldsymbol{a}_t))$, where $\boldsymbol{I} \in \mathbb{R}^{d \times d}$, and

$$\boldsymbol{\mu}_{t-1}(\boldsymbol{z}_t, \boldsymbol{a}_t) = [\mu_{t-1}(\boldsymbol{z}_t, \boldsymbol{a}_t, 1), \ldots, \mu_{t-1}(\boldsymbol{z}_t, \boldsymbol{a}_t, d)]^\top,$$
$$\boldsymbol{\sigma}_{t-1}(\boldsymbol{z}_t, \boldsymbol{a}_t) = diag(\boldsymbol{\Sigma}_{t-1}(\boldsymbol{z}_t, \boldsymbol{a}_t)) = [\sigma_{t-1}(\boldsymbol{z}_t, \boldsymbol{a}_t, 1), \ldots, \sigma_{t-1}(\boldsymbol{z}_t, \boldsymbol{a}_t, d)]^\top.$$

Our setup assumes that the components $\{\boldsymbol{x}_{t,l}\}_{l=0}^d$ are independent of each other given $\boldsymbol{z}_t, \boldsymbol{a}_t$. Therefore from Srinivas et al. (2010) we know that the mutual information for the $i$th GP model is:

$$I(\boldsymbol{x}_{1:T}, \boldsymbol{f}_{1:T}) = H(\boldsymbol{x}_{1:T}) - H(\boldsymbol{x}_{1:T} | \boldsymbol{f}_{1:T}) = \frac{1}{2} \sum_{t=1}^T \sum_{l=1}^d \ln\left( 1 + \frac{\boldsymbol{\sigma}_{t-1}^2(\boldsymbol{z}_t, \boldsymbol{a}_t, l)}{\rho^2} \right) \qquad (36)$$

because per component:

$$I(\boldsymbol{x}_{i,1:T,l}, \boldsymbol{f}_{i,1:T,l}) = \frac{1}{2} \sum_{t=1}^T \ln\left( 1 + \frac{\sigma_{t-1}^2(\boldsymbol{z}_t, \boldsymbol{a}_t, l)}{\rho^2} \right). \qquad (37)$$

Then, by the definition of maximum information gain $\gamma_{i,T,l}$ per node $i$ in the graph:

$$\gamma_{i,T,l} := \max_{A_i \subset \{\mathcal{Z}_i \times \mathcal{A}_i\}^T} I(\boldsymbol{x}_{i,1:T,l}, f_{i,1:T,l}) = \max_{A_i \subset \{\mathcal{Z}_i \times \mathcal{A}_i\}^T} \frac{1}{2} \sum_{t=1}^T \ln\left( 1 + \frac{\sigma_{i,t-1}^2(\boldsymbol{z}_t, \boldsymbol{a}_t, l)}{\rho^2} \right) \qquad (38)$$

Accordingly, we can write the maximum information gain between $\boldsymbol{x}_{i,1:T}$ and $\boldsymbol{f}_{i,1:T}$ as follows:

$$\gamma_{i,T} := \max_{A_i \subset \{\mathcal{Z}_i \times \mathcal{A}_i\}^T} I(\boldsymbol{x}_{i,1:T}, \boldsymbol{f}_{i,1:T}) = \max_{A_i \subset \{\mathcal{Z}_i \times \mathcal{A}_i\}^T} \frac{1}{2} \sum_{t=1}^T \sum_{l=1}^d \ln\left( 1 + \frac{\sigma_{i,t-1}^2(\boldsymbol{z}_t, \boldsymbol{a}_t, l)}{\rho^2} \right) \qquad (39)$$

$$\leq \sum_{l=1}^d \max_{A_i \subset \{\mathcal{Z}_i \times \mathcal{A}_i\}^T} \frac{1}{2} \sum_{t=1}^T \ln\left( 1 + \frac{\sigma_{i,t-1}^2(\boldsymbol{z}_t, \boldsymbol{a}_t, l)}{\rho^2} \right) = \sum_{l=1}^d \gamma_{i,T,l} \qquad (40)$$

***Note:*** *Bounds for $\gamma_{i,T,l}$ and $\gamma_{i,T}$.* Upper bounds on $\gamma_{i,T,l}$ are provided in Srinivas et al. (2010) for widely used kernels and scale sublinearly in $T$. We use $p_i = d |pa(i)|$ to represent the size

of the z-input to a GP. Recall that $q$ is the length of each action vector i.e. $\mathcal{A}_i \subset \mathbb{R}^q$. For the linear kernel $\gamma_{i,T,l} = \mathcal{O}((p_i + q) \log T)$, and for the squared exponential kernel $\gamma_{i,T,l} = \mathcal{O}((p_i + q)(\log T)^{p_i+q+1})$. We can use Eq. (39) to give bounds on $\gamma_{i,T}$ too e.g. if all GPs use independent linear kernels for each output component $\gamma_{i,T} = \mathcal{O}(d_i(p_i+q) \log T)$, and if all GPs use independent squared exponential kernels for each output component $\gamma_{i,T} = \mathcal{O}(d(p_i + q)(\log T)^{p_i+q+1})$.

**Step 2.** *A bound for model complexity* $\Gamma_T$.

Here we bound the model complexity Eq. (17)

$$\Gamma_T \leq \sum_{i=0}^{m} \frac{1}{\ln(1 + \rho_i^{-2})} \gamma_{i,T}. \tag{41}$$

For readability, in the following, we denote $\max_A(\cdot) := \max\limits_{\substack{A:\ A=\cup_i A_i: \\ \forall i:\ A_i \subset \{\mathcal{Z}_i \times \mathcal{A}_i\}^T}} (\cdot)$.

$$\Gamma_T = \max_{\{\mathcal{Z} \times \mathcal{A}\}^T} \sum_{t=1}^{T} \sum_{i=0}^{m} \|\boldsymbol{\sigma}_{i,t-1}(\boldsymbol{z}_i, \boldsymbol{a}_i)\|_2^2$$

$$\overset{\textcircled{1}}{\leq} \max_A \sum_{t=1}^{T} \sum_{i=0}^{m} \|\boldsymbol{\sigma}_{i,t-1}(\boldsymbol{z}_i, \boldsymbol{a}_i)\|_2^2 \overset{\textcircled{2}}{\leq} \sum_{i=0}^{m} \max_A \sum_{t=1}^{T} \|\boldsymbol{\sigma}_{i,t-1}(\boldsymbol{z}_i, \boldsymbol{a}_i)\|_2^2$$

$$\overset{\textcircled{3}}{\leq} \sum_{i=0}^{m} \max_{A_i} \sum_{t=1}^{T} \|\boldsymbol{\sigma}_{i,t-1}(\boldsymbol{z}_i, \boldsymbol{a}_i)\|_2^2 \overset{\textcircled{4}}{\leq} \sum_{i=0}^{m} \max_{A_i} \sum_{t=1}^{T} \sum_{l=1}^{d_i} \|\sigma_{i,(t-1)}(\boldsymbol{z}_i, \boldsymbol{a}_i, l)\|_2^2$$

$$\overset{\textcircled{5}}{\leq} \sum_{i=0}^{m} \frac{2}{\ln(1 + \rho_i^{-2})} \underbrace{\max_{A_i} \frac{1}{2} \sum_{t=1}^{T} \sum_{l=1}^{d_i} \ln\left(1 + \frac{\sigma_{i,(t-1)}(\boldsymbol{z}_i, \boldsymbol{a}_i, l)}{\rho_i^2}\right)}_{\text{maximum information gain Eq. (39)}}$$

$$\overset{\textcircled{6}}{=} \sum_{i=0}^{m} \frac{2}{\ln(1 + \rho_i^{-2})} \gamma_{i,T} \overset{\textcircled{7}}{=} \mathcal{O}(m\gamma_T),$$

where ①bounds $\mathcal{A}$ and $\mathcal{Z}$ with a box. ②is from the max over a sum. ③is due to the assumption of $\boldsymbol{x}_{i,t}$ being independent of $A_j$, $j \neq i$, conditioned on $A_i$. ④is due to Jensen's inequality. ⑤is due to the fact that for any $s^2 \in [0, \rho^{-2}]$ we can bound $s^2 \leq \frac{\rho^{-2}}{\ln(1+\rho^{-2})} \ln(1+s^2)$ Srinivas et al. (2010). This also holds for function $s^2(\cdot) := \rho_i^{-2} \sigma_{i,(t-1),l}^2(\cdot)$ since $\rho_i^{-2} \sigma_{i,(t-1),d}^2(\cdot) \leq \rho_i^{-2} k(\cdot, \cdot) \leq \rho_i^{-2}$ because $k_i(\cdot, \cdot) < 1 \; \forall i$ (bounded kernel assumption). ⑥is due to Eqs. (36) and (39). Finally, in ⑦we define $\gamma_T := \max_i \gamma_{i,T}$ being the maximum value of the maximum information gains over graph nodes.

Plugging this upper bound on $\Gamma_T$ into Theorem 2 completes the proof.

**Note:** *Sublinearity w.r.t. T of maximum information gain* $\gamma_T$.

Upper bounds on $\gamma_T$ will often scale sublinearly in $T$. This follows from $\gamma_{i,T}$ scaling sublinearly in $T$ for many popularly used kernels (see previous note). In particular, a linear kernel leads to $\gamma_T = \mathcal{O}(d(Kd + q) \log T)$ and a squared exponential kernel leads to $\gamma_T = \mathcal{O}(d(Kd + q)(\log T)^{Kd+q+1})$ (assuming output components are independent) since $\max_i p_i = Kd$.

$\square$

### A.2.4 DEPENDENCE OF $\beta_T$ ON $T$ FOR PARTICULAR KERNELS

Note that for Assumption 3 to hold under our RKHS assumptions, $\beta_T$ might depend on $T$. For a single observed variable corresponding to node $i$ at time $t$, for Assumption 3 to hold we must have $\beta_T = \tilde{\mathcal{O}}\left(\mathcal{B}_i + \frac{\rho_i}{d}\sqrt{\gamma_{i,t}}\right)$ (Chowdhury & Gopalan (2019) equation 9). For Assumption 3 to hold at

time $t$ for all $i$ we can apply a union bound and see that it is sufficient for $\beta_T = \tilde{\mathcal{O}}\left(\mathcal{B} + \frac{\rho}{d}\sqrt{\gamma_t}\right)$ where $\mathcal{B} = \max_i \mathcal{B}_i$ and $\rho = \max_i \rho_i$.

In the regret bound of Theorem 1, $\beta_T$ appears raised to the power $N$. For $\gamma_T$ corresponding to the linear kernel and squared exponential kernel, this will still lead to sublinear regret regardless of $N$ because $\gamma_T$ will be only logarithmic in $T$. However, for a Matérn kernel, where the best known bound $\gamma_T = \mathcal{O}\left(p(pT)^c log(pT)\right)$ with $0 < c < 1$, the cumulative regret bound will not be sublinear if $N$ and $c$ are sufficiently large. A similar phenomena with the Matérn kernel appears in the guarantees of Curi et al. (2020) which use GP models in model-based reinforcement learning.

## A.3    MAXIMIZING THE ACQUISITION FUNCTION

Our theoretical results assume access to an oracle that can maximize Eq. (10). Here we discuss how we approximate this oracle in practice.

In noiseless settings, instead of parameterizing each $\boldsymbol{\eta}_i$ as a neural network, we can parameterize it as a constant. This is because with no noise, the inputs to $\boldsymbol{\eta}_i$ ($\boldsymbol{z}_i$ and $\boldsymbol{a}_i$) given $\boldsymbol{a}$ are fixed. This keeps the space of parameters to optimize over small, meaning we can use an identical optimization procedure to that used in EIFN by Astudillo & Frazier (2021b) which is also an out-of-the-box optimizer in the BoTorch package Balandat et al. (2020).

For noisy settings where each $\boldsymbol{\eta}_i : \mathcal{A}_i \times \mathcal{Z}_i \to \mathbb{R}$ is a neural network, we use our own optimizer. For each initialization of $\eta$ parameters, we perform stochastic gradient descent to optimize both the $\boldsymbol{\eta}_i$ parameters and $\boldsymbol{a}$. We can do this because Eq. (12) is differentiable with respect to both $\boldsymbol{a}$ and the parameters of each $\boldsymbol{\eta}_i$. After running stochastic gradient descent on many random initializations we will have many solution candidates. We select the candidate with the highest acquisition function value. We use a large number of different random initializations because the acquisition function may be very non-convex. Other approaches, such as those considered in Curi et al. (2020) for model-based reinforcement learning, could also be adapted to optimize our acquisition function.

When parameterizing each $\boldsymbol{\eta}_i$ with a neural network, we always use a two layer feed-forward network with a ReLu non-linearity, To map the output into $[-1, 1]$ we put the output of the network through an element-wise Sigmoid.

In all noisy environments we estimate the expectation in the acquisition function (Eq. (12)) using a Monte Carlo estimate with 32 repeats for each gradient step. For noisy Dropwave we use 128 repeats because the environment is particularly noisy compared to other noisy environments.

## A.4    EXPERIMENTAL SETUP

Here we give additional notes to explain the details of our experimental setup.

The cross-validation for selecting $\beta$ is performed across $\beta = \{0.05, 0.5, 5\}$. For a given performance metric and task, we'll use the example of average reward on Dropwave, we select the $\beta$ that on average performs best across all other tasks for the same performance metric, called $\beta^*$. We then report the results in terms of average reward for running the algorithm (GP-UCB or MCBO) on Dropwave with $\beta = \beta^*$

For MCBO and EIFN, we use identical values for all shared hyperparameters (e.g. GP kernels) and use the original hyperparameters from Astudillo & Frazier (2021b). CBO methods do not have hyperparameters comparable with these methods because CBO methods do not model individual system observations (see Section 2.1). We run the original EICBO source code effectively unmodified Aglietti et al. (2020b).

For CBO tasks, if the agent selects no intervention (observational data) at a given timestep, we give the agent, for any algorithm, a single observational sample. This is different to in Aglietti et al. (2020b) where 20 observational samples are given.

There is no noisy version of Ackley because adding noise can make the environment unstable by producing very large or very small rewards. For all environments and all $X_i$, the noisy environment adds unit-Gaussian noise, except Dropwave where we scale this noise by $0.1$ to make the environment more stable.

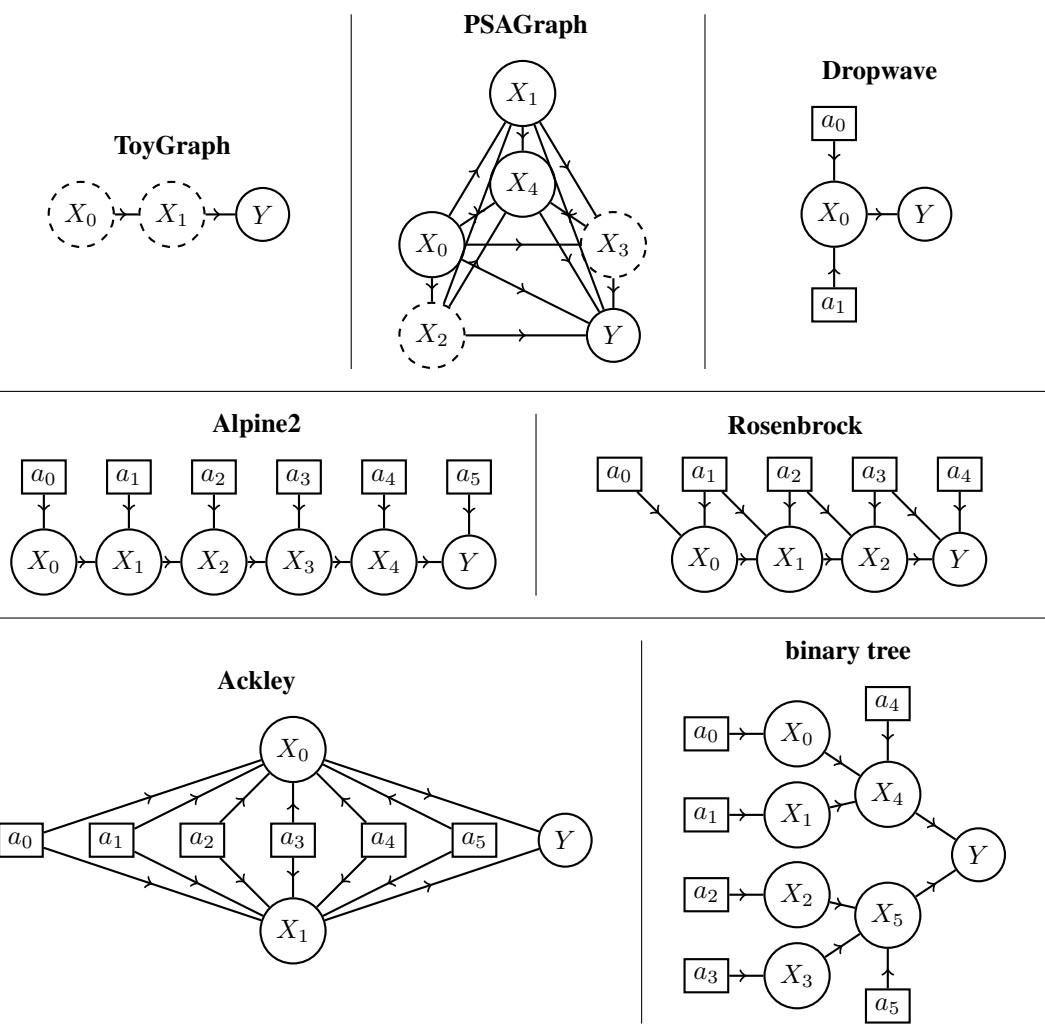

Figure 3: The DAGs corresponding to each task in the experiments, except binary tree which is used for illustrative purposes. Circles are observation or reward variables. Squares are actions in the function network setting. In hard intervention CBO, nodes with a dashed border can be intervened upon. All nodes represent a scalar random variable.

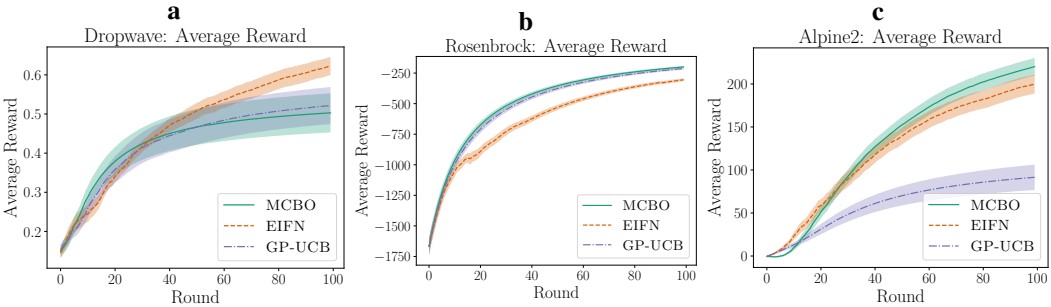

Figure 4: All average reward plots not used in the main paper. All settings are noiseless.

When applying MCBO to function networks we allow $\boldsymbol{a}_m$ to be nonzero, to match the function networks setting.

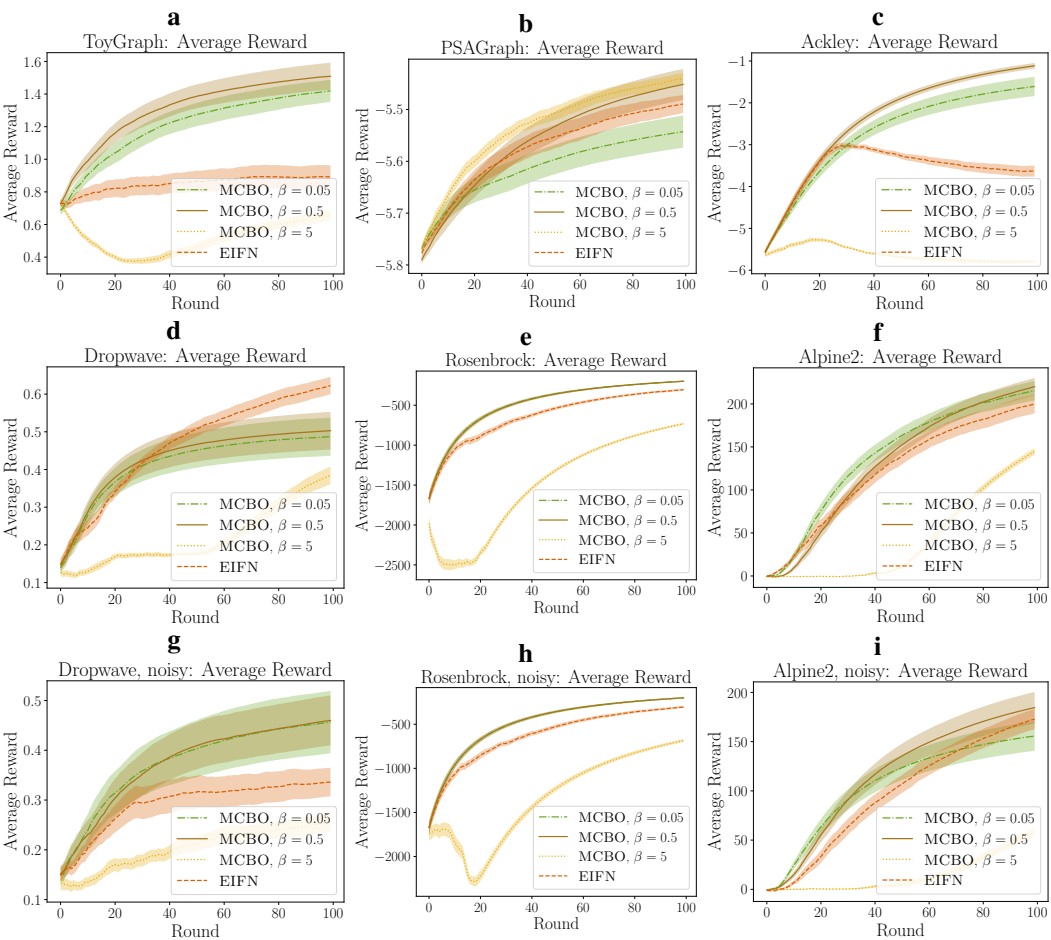

Figure 5: An ablation across $\beta$ where we plot average reward for all tasks.

## A.5 FURTHER EXPERIMENTAL ANALYSIS

**Average expected reward**

In Figure 4 we show the deterministic function networks plots not included in the main paper. In Figure 5 we show the average reward for all $\beta$ considered in the cross-validation.

**Best expected reward** Here we also report the best expected reward as a function of the number of rounds $T$. The best expected reward at time $T$ is defined by

$$\max_{\boldsymbol{a} \in \{\boldsymbol{a}_t\}_{t=0}^T} \mathbb{E}_\omega[Y \mid \boldsymbol{a}].$$

This is similar to but not directly inversely related to simple regret. Simple regret algorithms require an inference procedure to select a final action after $T$ rounds of exploration. This procedure could be computationally expensive for the CBO setting. For example, a reasonable choice would be to report a final action based upon maximizing a lower confidence bound of the objective in Eq. (4), however a closed form expression for this does not exist for CBO. Best expected reward instead assumes access to an oracle that computes the highest expected reward of any action the algorithm has previously chosen. Our plotting of the best expected reward is consistent with previous work on CBO (Aglietti et al., 2020b).

When performing the same cross-validation procedure as for the average expected reward case, we find that the performance of MCBO varies drastically across tasks. This is shown in Figure 6. Selecting $\beta$ is known to be a difficulty with UCB-based method (Merrill et al., 2021). Figure 7 shows the performance in terms of best reward for all 3 possibilities for $\beta$. When selecting just

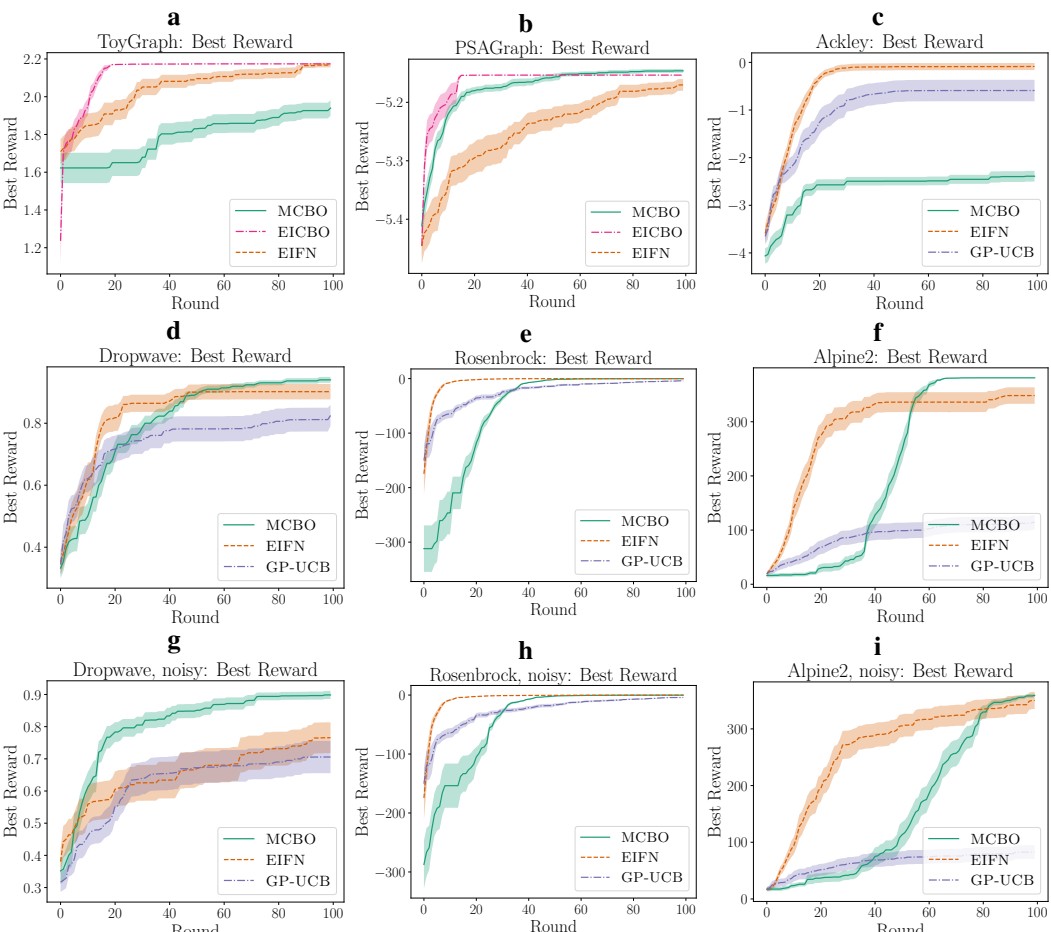

Figure 6: All best reward plots.

the best $\beta$ among the 3 tried, we find that MCBO is very strong on all tasks (with the exception of best reward on ToyGraph, where it underperforms the baselines). The optimal $\beta$ can vary by an order of magnitude or more across tasks in our case. This is likely because all of our settings have very different functional relationships and graph structures. As discussed in Section 4, this can be understood from our cumulative regret guarantee in Theorem 1.

The plots of average and best scores for other values of $\beta$ (Figs. 5 and 7) suggest that $\beta$ can be made larger for MCBO to trade-off exploration vs. exploitation depending on whether simple or cumulative regret is more of a priority. Meanwhile, expected improvement methods are focused almost almost entirely on exploration.

**Runtimes** In Figure 8 we show the total runtime of each method for all 100 rounds. EIFN and MCBO run on equivalent hardware, which has 4 times more ram than the hardware used for GP-UCB and EICBO. MCBO generally has a much longer runtime than EIFN in noisy settings where the $\eta_i$ are parameterized by neural networks. Generally in BO, because the unknown function is often assumed expensive to evaluate, we are less concerned about the time required to optimize the acquisition function and more concerned with notions of statistical efficiency such as regret. Our implementation for noisy settings could likely be sped-up with more parallelism to make it more comparable to EIFN runtimes. In noiseless settings, where the optimization methods used are roughly equivalent between EIFN and MCBO, the two methods have comparable runtimes.

**Non-monotonic regret of EICBO** In Fig. 2 (a,b) we found that EICBO had a non-monotonic average reward, which could likely be explained by the use of an expected improvement acquisition function. We decided to clarify that the non-monotonic average reward was because of the algorithm and not because of differences in our setup compared to Aglietti et al. (2020b). Besides us evaluat-

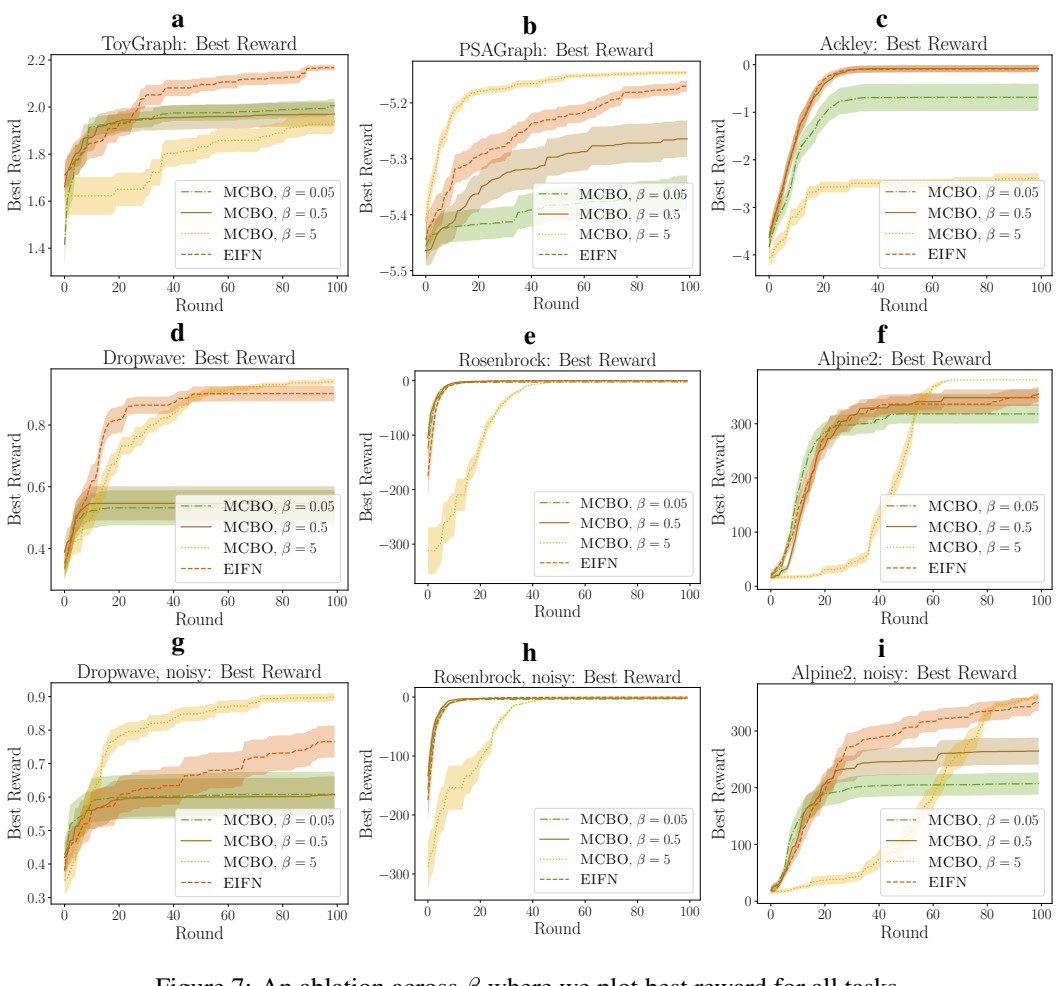

Figure 7: An ablation across $\beta$ where we plot best reward for all tasks.

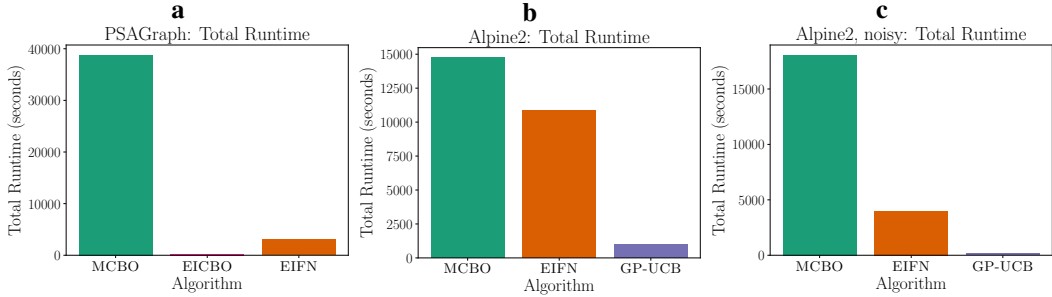

Figure 8: Runtimes of experiments for a **a**) CBO setting, **b**) noiseless function network setting, and **c**) noisy function network setting.

ing in terms of average reward instead of best reward, compared to Aglietti et al. (2020b) we also used smaller initial sample sizes (random samples the agent obtains before $t = 0$) of observational data. This was to make the setting more challenging, since we found that with extra observational data ToyGraph and PSAGraph could be solved with very few samples. When reproducing the experiments of Fig. 2 (a, b) with additional starting observational data, we found the same qualitative results (non-montonicity) for the average reward case. Note that this result is not inconsistent at all with the experimental results of Aglietti et al. (2020b), since they only evaluate in terms of best reward.

