# OpenReview forum: "Model-based Causal Bayesian Optimization"
_ICLR.cc/2023/Conference — ICLR 2023 notable top 25%_

### Official Review · Reviewer_B1mp · 2022-10-18

**Confidence:** 3
**Correctness:** 4
**Technical Novelty And Significance:** 3
**Empirical Novelty And Significance:** 3
**Recommendation:** 8

**Clarity, Quality, Novelty And Reproducibility:**

Quality:
Other than the technical issues raised in the previous section, the paper is generally well-written. The experiments section is good.

Clarity:
The paper suffers from a lack of clarity in some technical parts as raised in point 2 of the previous section.

Originality:
The paper is reasonably novel as it provides theoretical guarantees for a setting in which previous work did not provide such guarantees.


**Strength And Weaknesses:**

Strengths:
1. The problem of causal BO is motivated well and explained clearly with diagrams.
2. The algorithm's performance is empirically well-supported.

Weaknesses:
1. One concern comes from Assumption 3 not providing an explicit formula for the sequence $\beta_t$. The authors claim that there is an "overall sublinear regret for MCBO". However, $\beta_T^N$ appears in the regret bound, where $N$ is the maximum distance from a root node to the reward node. Without an explicit formula for $\beta_t$ (as is usual in this line of work), there is no reason to believe that  $\beta_T^N < \mathcal O(\sqrt{T})$. Even if we assume that the usual formula for $\beta_t$ (Chowdhury and Gopalan, 2017) is applicable, we still have that $\beta_t = \mathcal O(\sqrt{\gamma_{t-1}})$. For the Matern kernel, $\gamma_T = \mathcal O(T^{d(d+1)/(2\nu + d(d+1))} \log T)$ (Srinivas et. al., 2010). Thus, if the Matern kernel was used, there would be some value of $N$ beyond which the regret bound is not sublinear. Assumption 3 needs to be even stronger than it currently is for the sublinearity claim to be true. Ideally, the authors could provide an explicit formula for $\beta_t$ instead of assuming such a sequence exists, or at least discuss why constructing such a sequence would be hard.

2. Each $f_i$ is a vector-valued function modelled by a vector-valued GP. However, the vector-valued GP as presented in this paper does not match with the usual understanding of a vector-valued GP. From Sec. 3 in Alvarez et. al. (2012) (https://arxiv.org/pdf/1106.6251.pdf), vector-valued GPs have matrix-valued kernel functions $\mathbf K: \mathcal S \times \mathcal S \rightarrow \mathbb R^{d \times d}$. However, this paper's regularity assumptions use a vector-valued kernel function $k_i: \mathcal S \times \mathcal S \rightarrow \mathbb R^{d_i}$. The definition of GP posterior mean and variance Equs. 7 and 8 are not well-defined as they rely on $\mathbf K_t$ where $(\mathbf K_t)_{ij} \coloneqq k_i(\mathbf z_i, \mathbf a_i; \mathbf z_j, \mathbf a_j)$, but $k_i$ is vector-valued according to their definition, so $\mathbf K_t$ is not a matrix. Can the authors clear up this confusion?

3. Why is UCB not tested on ToyGraph and PSAGraph?

4. Other issues:
- For the definitions of $\mathcal X$ and $\mathcal Z_i$, did you mean to use Cartesian product $\times$ instead of union $\cup$?
- The legends of the plots say MBCO instead of MCBO.

**Summary Of The Paper:**

The paper proposes a new algorithm for causal Bayesian optimization for both hard and soft interventions with theoretical guarantees. The authors use a reparameterization technique to optimize the acquisition function which is a maximum over functions. They empirically show that the algorithm mostly outperforms relevant baselines.

**Summary Of The Review:**

I believe this paper has potential. Theoretical guarantees are important and the experiments show that their algorithm performs well in comparison to baselines. However, unless I am mistaken, the paper suffers from non-trivial technical issues. Assumption 3 as presented is already very strong but actually needs to be even stronger to properly achieve a sublinear regret bound. The definition of the GP posterior mean and variance is not well-defined and hinders understanding of their method. Until these issues are addressed, I am hesitant to recommend acceptance.

POST-REBUTTAL UPDATE: The authors have satisfactorily answered the main technical concerns and have improved the paper. I have improved my score to recommend acceptance.

---

> ### Author Response · Authors · 2022-11-11
> **Response to the original comment of reviewer B1mp**
>
> We would like to thank the reviewer for their constructive feedback. We have incorporated the suggestions into the revised paper and respond to each comment individually below:
>
> (**Dependence of $\beta_T$ on $T$**) The revised paper includes a discussion on how $\beta_T$ scales with $T$ (see appendix A.2.4, also referenced in the main body) . We  agree with the reviewer on the nature of the resulting bound when the matern kernel is used, where sufficiently large N results in the regret bound no longer being sublinear. A similar result for the matern kernel occurs in the work we cite on model-based reinforcement learning by Curi et al. (appendix H.2 discussion of Theorem 3).  For many other simpler kernels the dependence of $\beta_T$ on T is such that accounting for this only introduces log terms, meaning the overall cumulative regret is still sublinear.
>
> (**Vector-valued GP**) We thank the reviewer for the note, it was indeed scalar output GPs written in the paragraph on modeling with GPs. In the revised version we have adapted it to model vector-output functions as follows: by using a single GP that takes in the index of the output dimension under consideration as an input. Importantly, the proofs already relied on this formulation of vector-valued GPs, so they stay unchanged.
>
> (**Use of UCB in the CBO settings**) GP-UCB was not evaluated on the hard intervention CBO setups because there one must select both a set of intervention targets (discrete) and the values those targets will be fixed to (continuous). GP-UCB models the reward given actions with a single GP that has fixed input dimensionality. However, for hard intervention CBO the continuous action input space will vary depending on how many intervention targets are selected. Therefore GP-UCB would have to be modified to use a different GP model for each possible set of intervention targets, which means it cannot generalize across intervention targets like other methods. This phenomenon is explored for another classical BO algorithm (expected improvement) in Aglietti et al. (2019). If the reviewer feels strongly that GP-UCB should be compared to somehow in the hard intervention CBO setting, we are open to reconsidering our choice and including such experiments in the final version.

---

> > ### Comment · Reviewer_B1mp · 2022-11-17
> > **Response**
> >
> > Thank you for your response to my questions. I will update my review accordingly.

---

### Official Review · Reviewer_UVNu · 2022-10-22

**Confidence:** 3
**Correctness:** 4
**Technical Novelty And Significance:** 3
**Empirical Novelty And Significance:** 3
**Recommendation:** 8

**Clarity, Quality, Novelty And Reproducibility:**

The paper is clear both in terms of theoretical proofs and empirical guarantees.
Furthermore, the notation is consistent while necessarily complex.
The assumptions on which the contributions are based are listed and discussed in a clear manner.
The same applies to the discussions about limitations and possible solutions to overcome them.
Formulas, pseudocodes and images are of good quality and help the interested reader to go through the work.
I vote for the experimental layout (setting) is bith correct and reproducible, in the light that the author/s make available the software code. However, I had not time to check it myself.
The examples are meaningful, in adequate quantity and increasing complexity.
Finally, I think that the degree of novelty of the paper is good and what developed and describe is relevant for the target field.

**Strength And Weaknesses:**

Strength points:
    - Model-based approach over structural causal models that inherits the theoretical framework for causal inference.
    - Guarantees in terms of bounded cumulative regret.
    - Assumptions make sense, allowing to generalize to a wide range of settings.

Weak points:
    - No unobserved confounding could be a naive assumption.
    - Knowing the true causal graph combined with causal sufficiency means to know the causal mechanism, even if the functions are unknown.

**Summary Of The Paper:**

The paper is about Bayesian Optimization (BO), and, in particular, Causal Bayesian Optimization (CBO) based on a novel model-based approach. Authors propose to extend the existing CBO using a model-based approach, along with proofs for bounding the cumulative regret and more realistic assumptions compared to current CBO approaches. The reference model is the Structural Causal Model (SCM) with known casual graph but unknown functional relations, which enables the representation of both observational and interventional distributions. Authors explore both soft and hard interventions, with and without constraints over actions. The objective is to select a sequence of actions to maximize the expected reward by minimizing the cumulative regret function with a finite horizon. Since the acquisition function cannot be evaluated in closed form, authors apply the "kernel trick" and leverage gradient-based optimizers. Finally, both theoretical proofs and empirical evaluations are provided.

**Summary Of The Review:**

The paper is well structure, organized and written and I enjoyed reading it.
Theoretical statements are presented and supported by the associated proofs .
Furthermore, empirical achievements are well supported a rich set of numerical experiments and obviously the associated results which are presented and discussed in an effective manner.
The presented approach is novel for the considered research area and its results are promising also when taking into consideration the assumptions made by the author/s.
It is also worth to mentin that the topic of the paper is relevant and discussed in details, thus representing an added value for the venue.
In conclusion I think the paper deserves consideration for being presented to a restigious venue as ICLR.

---

> ### Author Response · Authors · 2022-11-11
> **Response to the original comment of reviewer UVNu**
>
> We thank the reviewer for their feedback.
>
> Regarding unobserved confounding, in the future we hope to explore how ideas from MCBO might be applied to settings where causal sufficiency cannot be assumed. We believe that before then there are still many exciting uses for MCBO in settings that assume causal sufficiency, as also suggested by the work of Astudillo and Frazier (2021).

---

> > ### Comment · Reviewer_UVNu · 2022-11-12
> > **I agree with you**
> >
> > I agree with you, there is still a lot to be done in the direction of limiting causal sufficiency assumption.

---

### Official Review · Reviewer_wHmj · 2022-10-24

**Confidence:** 3
**Correctness:** 2
**Technical Novelty And Significance:** 2
**Empirical Novelty And Significance:** 2
**Recommendation:** 6

**Clarity, Quality, Novelty And Reproducibility:**

# Clarity

The paper is well-written and mathematically rigorous.

# Quality

A high-quality paper.

# Novelty

The novelty is low. There are sections that, if expanded, will be far more impactful and interesting.

# Reproducibility

The authors have included the code. I have not gone through it.


**Strength And Weaknesses:**

This review proceeds section by section per your paper.

# Abstract

- It is confusing when you say "downstream" variable of interest. Certainly BO is a sequential method but you are targeting static causal diagrams wherein which "downstream" makes it sound as if you mean something in the future. Looking at your graphs you are targeting the root node of each CD (as per causal inference literature, not decision tree root-nodes who flip the terminology -- though I notice you are calling the outcome variable the 'sink').
- CBO, as introduced by Aglietti et al., target a full SCM not just a set of SEMs as you infer. Unless you are introducing a new method with the same name, then you are not discussing the same model framework.
- The original CBO paper does not assume noisless measurements. Where does it say that? The implementation of the model CBO (see here for the complete graph from the paper: https://github.com/VirgiAgl/CausalBayesianOptimization/blob/master/graphs/CompleteGraph.py) clearly has an epsilon parameter (the noise parameter for the observational samples). Your statement is either wrong or you are referring to something different.
- BO does not come with guarantees either so I am not sure how yours is a valid point? BO is not guaranteed to converge and so by extension neither is CBO.

# Introduction

- You should mention that this idea was first proposed by Aglietti et al. and yours is not a new idea w.r.t. the second paragraph. Similar ideas were introduced by the Lattimore brothers in their 2016 paper on causal bandits and Bareinboim in their 2015 before them. Optimal decision making + causal knowledge is not a new idea.
- I am curious as to why you depart from standard ideas of representing observationa and interventions in graphical form w.r.t. figure 1. It is standard to make interventional nodes V as upper-case labels and then repsent interventions as removing any incoming arcs to that node which is relabelled with small-caps label v, Hence, why have you departed from this formalism? What is the reasoning?

# BACKGROUND AND PROBLEM STATEMENT

- You are assuming causal sufficiency (no UCs) but the original CBO papers does not. Is that not a bit of a limiting assumption? It certainly limits the type of problems that you can deal with.
- It would be clearer if you used more set-notation e.g. to refer to all variables bar the outcome variables write $\mathbf{X} \setminus \{Y\}$. It is clearer.
- The same thing for graph terminology: use standard family relationships to refer to e.g. the parents of X_i i.e. pa(X_i) if you are not including also the argument and Pa(X_i) if you are. Please stick to this and don't introduce new notation.
- Can you please provide the pagenumber for "soft intervention with unknown effect" in Pearl's book, I cannot seem to find it.
- I do not follow why graph mutilation (i.e. hard interventions) is an issue w.r.t. your method? Please explain. I do not follow the logic or rather reasoning for going with soft or shift interventions.
- Not intervening with a hard-intervention would simply be to observe the system which results in no grap mutilation which means the DAG is not modified at all. This seems to be a odds with your previous point.
- When we cannot intervene on a node we usually call it a "non-manipulative variable" - this is well studied.
- As to _what_ we should intervene on, the intervention set as it is called, is also very well studied. See Lee and Bareinboim's 2018 paper on 'where to intervene' - they prove specifically that when there are no UCs present (like you are assuming) it is always best to intervene on _all_ the parents of the target variable.
- You ought to reference the metric that you are using as it is not novel and has been used for decades to evaluate bandits.
- Now in equation 6 you are using hard interventions but which in your setting will not mutilate graph is what you are saying?
- I find it strange that you have not discussed _what_ you plan to intervene on given that you'll be exploring the whole powerset space of interventions $\mathcal{P}(\mathbf{x} \setminus \{Y\})$ and this seems like a detail worth exploring, if not this will be terribly inefficient.

# ALGORITHM

- Where does equation 9 come from?
- Please explain the last sentence of this paragraph where you say: "In the following, we set βi,t = βt for all i such that the confidence bounds are still valid."
- After equation 10 then it sounds as if you are indeed exploring the whole powerset over intervention variables. That is a _very_ large set to explore particularly as you do not have UCs so the best intervention is always the parents of Y.
- What is $\mathcal{I}$?
- In the hard intervention paragraph you are saying that use the MIS/POMIS ideas of Lee and Bareinboim but you're not really; you are using one of their definitions (number 1) to find the minimal intervention sets because, again, you do not have UCs, their work is barely applicable to your setting. Their ideas come into their own when there are UCs but otherwise, with chain-structured graphs you just remove intervention sets which topologically yield the same outcome expression.

# THEORETICAL ANALYSIS

- This is a very nice section, nicely done.
- It would have been nice to have a few examples of the bound in theorem 1 (give it an equation number so it can be referenced) applied to a few of the examples in figure 2.
- Why are the a's in the figures no longer bold as in figure 1(b)?
- I think you are taking too many liberties with how you are drawing your graphs. Typically, in causal inference, we reserve dashed edges for items that regard unobserved confounders (mind you, Aglietti et al. also takes liberty with their graph drawing so mine is a general comment).
- You should look into dynamic treatment regimes; a lot of your examples graphs have the same topological structure (and ordering). Could be interesting to see if your ideas are applicable there.

# EXPERIMENTS

- It would seem odd for you to call the original paper 'EICBO' and not 'CBO' - call it the latter, as they were first to publish this idea.
- It is very strange that you do not deploy all methods on all test cases in figure 3.

**Summary Of The Paper:**

The authors present a version of the CBO paper from 2020. They introduce some new minor ideas of how to fuse BO with causal inference ideas.

**Summary Of The Review:**

This is a twist on an earlier idea. I do not think there is enough novelty to warrant inclusion as it stands.

---

> ### Author Response · Authors · 2022-11-11
> **Reply to the original comment of reviewer wHmj. Part 1**
>
> We would like to thank the reviewer for the detailed feedback. We divide our response here into two comments. In the first, we address major concerns regarding the paper’s novelty and related work on CBO. In the second, we address other feedback. We have revised the paper based on the suggestions and encourage the reviewer to follow-up if there are further concerns.
>
> (**Novelty**) Optimal decision making and causal knowledge is indeed not a new idea and as we state in the related work, the CBO paper was the first to introduce the causal setup to BO. There are, however, several significant differences between CBO and MCBO which makes the contributions of each interesting in their own right.
>
> First, MCBO has significant methodological differences compared to CBO. MCBO learns the full system model, while CBO uses the do-calculus to estimate causal effects.
>
> Second, the CBO paper provides no guarantees. The algorithm proposed in CBO combines expected improvement and an epsilon greedy policy to give a heuristic approach. We provide guarantees for our approach in the soft intervention setting (the CBO paper only considers the hard intervention setting) and provide an interpretable regret bound.  With our approach we can interpret the regret bound to understand how the performance of the algorithm might be affected by the graph topology, GP kernel and system noise. No such theory is present for CBO.
>
> Another important difference is that MCBO can handle noisy observation samples. We give more details on this in the next section.
> Finally, in the empirical evaluation, we show that MCBO outperforms CBO under a cumulative regret performance metric.
>
> The reviewer has asked us to mention that CBO was first introduced in [3] in our introduction. We however intentionally avoid it for the sake of clarity: our setup is not the same as the setup in [3]. For example, we assume causal sufficiency but also allow for soft interventions. We believe there is no issue with appropriate attribution. We clearly introduce the setting we consider (citing [3] when we do this) before explicitly stating in the related work that [3] were the first to introduce the CBO setting, clarifying the ways in which their setup differs from ours.
>
> (**The noiseless interventional data assumption in CBO**) This assumption is not explicitly stated in the CBO paper but is made implicitly in the experiments. While observational samples are indeed noisy ([2] line 71 inside “observe”, where they query a noisy observation from a batch of noisy observational samples precomputed before the experiment is run),   the interventional samples are noiseless since CBO directly queries the expected value of the target given the intervention ([2] line 43 inside “Intervention_function” they average across 100000 intervention samples). Classical EI-based BO also provides no guarantees when rewards are noisy (see [1] section 2.1 for an overview). In our experiments, we still allow CBO to obtain noiseless samples following the original CBO paper and implementation, but MCBO obtains noisy observations.
>
> **Q**:  “The novelty is low. There are sections that, if expanded, will be far more impactful and interesting.”
>
> **A**:  We could not find from the rest of the review which sections the reviewer would want to be expanded (and how) that could improve the impactfulness. Please clarify and we will follow-up.
>
> **Q**:   “You are assuming causal sufficiency (no UCs) but the original CBO papers does not. Is that not a bit of a limiting assumption? It certainly limits the type of problems that you can deal with.”
>
> **A**:  This is an important difference which we make clear in the related work and we think leads to the contributions of CBO and MCBO being very different. The original CBO allows for one to model systems that even have unobserved confounding, however this is with the caveat that the method has no theoretical guarantees, uses only hard interventions, needs noiseless interventional samples, and can only handle certain kinds of unobserved confounders (graphs where the do-calculus can be applied). We assume no unobserved confounding, so consider a smaller class of problems, but can provide theoretical guarantees, consider a very general class of soft interventions and allow for noisy interventional samples. We think that the setting of causal sufficiency still allows for modeling interesting problems, as also suggested by the work we cite by Astudillo and Frazier [4] on the related function networks setting.
>
> [1] Letham, Benjamin, et al. "Constrained Bayesian optimization with noisy experiments" Bayesian Analysis, 2019
>
> [2] https://github.com/VirgiAgl/CausalBayesianOptimization/blob/master/CBO.py
>
> [3] Virginia Aglietti, et al  “Causal Bayesian Optimization” AISTATS, 2020b.
>
> [4] Astudillo, Raul, and Peter Frazier. "Bayesian optimization of function networks." NeurIPS, 2021

---

> > ### Author Response · Authors · 2022-11-11
> > **Reply to the original comment of reviewer wHmj. Part 2**
> >
> > **Response part 2**
> >
> > **Guarantees**
> >
> > **Q**: BO does not come with guarantees either so I am not sure how yours is a valid point? BO is not guaranteed to converge and so by extension neither is CBO.
> >
> > **A**: We would like to ask the reviewer to clarify the statement ‘BO does not come with guarantees’ and what point this invalidates. There are many BO algorithms cited in our work (Srinivas et al. 2011, Astudillo and Frazier 2021, Kusakawa et al. 2021) that come with guarantees and have been of interest to the community. The provided guarantees are an important part of the MCBO approach.
> >
> > **Soft vs hard interventions: where to intervene**
> >
> > **Q**: “I do not follow the logic or rather reasoning for going with soft or shift interventions.”
> >
> > **A**:  The theory handles soft interventions with potentially unknown effect. The shift intervention is given as an example of a soft intervention but our result is much more general.
> >
> > **Q**:  “it is always best to intervene on all the parents of the target variable”
> >
> > **A**: This is true when you have do-interventions and no constraints on what you can intervene upon. In our setting we have soft interventions (not studied to our knowledge in the work you referenced) and possible constraints that don’t allow for intervening on all parents of the reward anyway.
> >
> > **Q**:  “I find it strange that you have not discussed what you plan to intervene on given that you'll be exploring the whole powerset space of interventions P(x∖Y) and this seems like a detail worth exploring, if not this will be terribly inefficient.”
> >
> > **A**: For soft interventions if there are no constraints on the set of intervention targets, it can be best to intervene on every node.
> > For hard interventions we already state that we use the method of Lee and Bareinboim [5] in order to prune the feasible intervention set.
> > There is a section in the setup already discussing constraints, so please note that in our setup the space of interventions does not necessarily need to include the powerset of all nodes. We hope that our revised paper now makes this clearer.
> >
> > **Q**:  “In the hard intervention paragraph you are saying that use the MIS/POMIS ideas of Lee and Bareinboim but you're not really”
> >
> > **A**: We have reworded the relevant section to make it clearer that we use the minimal intervention set definition to reduce the search space, but not the full algorithm proposed in [5].
> >
> > **Q**:  “I do not follow why graph mutilation (i.e. hard interventions) is an issue w.r.t. your method? Please explain.”
> >
> > **A**: Empirically it is not an issue at all. We rewrite our algorithm for hard interventions in Algorithm 2 and use this in the relevant experiments. For the guarantees in the theoretical section, we consider soft interventions (under some additional technical assumptions: assumptions 1,2,3) only because hard interventions would always break Assumption 1 (that all functions in the system are continuous with respect to the parent variables and action inputs). Hard interventions would break this because we can cause a large change in the child variable by going from intervening to not intervening on it.
> >
> > **Q**:  “It is very strange that you do not deploy all methods on all test cases in figure 3.”
> > We want to note that CBO is not directly applicable to function networks (similar to soft interventions), that is why it is not deployed in this setting.
> >
> > **Algorithm**
> >
> >
> > **Q**:  “Where does equation 9 come from?”
> >
> > **A**: Equation 9 is the definition of the set of feasible models. It is a standard tool in UCB-based Bayesian optimization approaches. Two citations are given in the paragraph below the equation.
> >
> > **Q**:  “Please explain the last sentence of this paragraph where you say: "In the following, we set $β_{i,t} = β_t$ for all i such that the confidence bounds are still valid."”
> >
> > **A**: For each confidence bound in $\mathcal M_t$, $\beta_{i, t}$ needs to be large enough so that the confidence bound holds. To simplify notation, for each $t$ we pick a single $\beta_t$ that is large enough so that the bound corresponding to every $\beta_{i, t}$ holds when we set $\beta_{i, t} = \beta_t$.
> >
> > **Q**: “I am curious as to why you depart from standard ideas of representing observationa and interventions in graphical form w.r.t. figure 1”
> >
> > **A**: For soft interventions our method is optimizing over a set of actions that modify the intervention target without breaking the connection to the direct causal parents. We found it clearer to include these actions directly in the graph (as parents of the variables they affect). For soft interventions, there is no breakage of edges in the causal graph so we don’t remove incoming arcs to nodes in the way the reviewer describes.
> >
> > Please follow-up with us on this clarification with any further concerns and let us know if it clarifies your existing concerns regarding novelty.
> >
> > [5] Sanghack Lee and Elias Bareinboim. “Structural causal bandits with non-manipulable variables”. AAAI, 2019.

---

### Official Review · Reviewer_geFQ · 2022-10-29

**Confidence:** 4
**Correctness:** 3
**Technical Novelty And Significance:** 3
**Empirical Novelty And Significance:** 2
**Recommendation:** 6

**Clarity, Quality, Novelty And Reproducibility:**

This paper is very well written, and the code used in the empirical evaluation is publicly available. Thus, this work performs well in terms of clarity and reproducibility. The proposed method is technically sound, thus making this paper perform well in terms of quality. As stated above, my main concern is that the discussion regarding novelty over prior work is misleading. It is also somewhat disappointing that realistic experiments were not considered in the empirical evaluation.

**Strength And Weaknesses:**

Strengths:
1. This work considers a relevant problem setting and proposes a technically sound method with good empirical performance and theoretical guarantees.
2. This paper is very well written overall.
3. An implementation of the code used in the empirical evaluation is publicly available.

Weaknesses:
1. Novelty: My main concern is related to this work's novelty claims. I believe the problem setting considered by this work is the same as that of Astudillo and Frazier (2021) in the sense that a function network could be written as a causal graph and vice versa. Both papers consider a directed acyclic graph where nodes in the graph take as input an action variable and the outputs of their parent nodes. I would like the authors to clarify what they mean in the following statement: "Moreover, function networks are not causal models and do not model actions as interventions." Is it only the use of the causal model formalism that makes these two problem settings different? Assuming this is the case, the primary source of the novelty of this work becomes the proposed acquisition function. I believe this is enough for this work to warrant publication, but the authors should clarify this. The authors should also explain that Kusakawa et al. (2021) also developed a UCB-based acquisition function that comes with similar sublinear regret guarantees in the cascade setting. How does the acquisition function proposed in this work compare with that proposed by Kusakawa et a. (2021) for causal graphs with a cascade structure?
2. Lack of realistic experiments: As discussed by the authors, the problem setting considered in this work arises in many critical real-world applications. However, they only considered one realistic test problem in their empirical evaluation. Moreover, this problem does not seem to be built upon a high-fidelity simulator or real-world data.

Other minor comments:
1. The authors should discuss how $\eta$ is modeled in more detail. I wonder how the choice of the neural network architecture affects performance.
2. This work falls within an existing line of research on grey-box Bayesian optimization (see, e.g., Astudillo and Frazier 2022). I recommend the authors discuss this literature.

Astudillo, R., & Frazier, P. I. (2022). Thinking inside the box: A tutorial on grey-box Bayesian optimization.

**Summary Of The Paper:**

This paper considers Bayesian optimization over objective functions defined via a causal graph, where the nodes correspond to functions defining the relationships between nodes in the graph (including the action variables). Within this framework, a UCB-based acquisition function is proposed. This acquisition function is shown to enjoy sublinear regret. Moreover, it is shown to perform competitively with respect to several state-of-the-art methods across several test problems.

**Summary Of The Review:**

This paper proposes a novel and technically sound acquisition function for causal BO. This acquisition function performs competitively in numerical experiments. My main concerns are that (1) this work's discussion regarding novelty over prior work is somewhat misleading, and (2) realistic experiments were not considered. Were these two aspects of the paper improved, I would support accepting this work.

---

> ### Author Response · Authors · 2022-11-11
> **Reply to the original comment of reviewer geFQ**
>
> We thank the reviewer for the careful reading and assessment of our work as well as the suggestions. We incorporate the suggestions in the revised paper and further respond to the questions and concerns individually below:
>
> (**Novelty w.r.t FNBO**) Astudillo and Frazier (2021) introduce the important setup of Function networks (FNs) and it is similar to the MCBO setup but in a special case only. The MCBO setup generalizes FNs to a richer class of problems in the following ways.
> First, FN can indeed be written using the formalism of a causal model, however the class of modeled interventions (that are equivalent to the *actions* in FNs) is constrained to *soft interventions only*. FNs do not consider a discrete set of intervention targets and do not also allow for modeling actions as do-interventions. That means that in the case of hard interventions it is not possible to equivalently write FNs as in Eq. 6 and vice versa. Our setup can be seen as combining the hard (Aglietti et al. 2020) and soft (Astudillo and Frazier 2021) interventions setups.
>
> Second, even if one only considered the soft intervention (or FN) setting, MCBO contains several novelties not present in the work of Astudillo and Frazier. First, our setting incorporates system noise. This is not present in the FNs setup. The noiseless setting is necessary for the guarantees of EIFN to hold, since it is based upon the idea of expected improvement. MCBO introduces a setup where all observations can be noisy and our theoretical guarantees hold in such a setting. Moveover, the theoretical guarantees of MCBO are stronger because we bound the cumulative regret, while the results in Astudillo and Frazier show only convergence. Finally, MCBO has favorable empirical performance in our experiments in particular in terms of cumulative regret. Therefore we think the source of novelty in this work comes not only from having considered a setup allowing for hard and soft interventions, but even more so from the handling of system noise, stronger theoretical guarantees, and favorable empirical results. We elaborate on these differences in the final paper version and rewrite the confusing sentence of "FNs are not causal models and do not model actions as interventions" by a statement explaining that the causal model formalism allows for hard interventions while FNs do not.
>
> (**Difference to Kusakawa et al. (2021)**) Kusakawa et al. (2021) indeed develops a UCB-based acquisition function for the chain graph setting. The main focus of the paper is on two other methods, namely EI and credible intervals, and their UCB-based method is in the appendix and is not included in any experiments. Algorithm-wise, Kuwasaka et al. compute a closed form confidence bound using Lipschitz continuity assumptions (Theorem B.1), that could be very loose. In contrast, we provide a practical method that uses a reparameterization trick to compute our upper confidence bound objective. Similar to MCBO, their method is designed for noisy systems, however the DAG is constrained to a cascade type only. In the revised paper, we have added more detail on the comparison.
>
>
> (**Experiments**) While our experiments are all on toy or synthetic experiments, we feel that we have covered the range of experiments available to us given prior work. We agree that there is a need for CBO benchmarks that are based on more realistic simulators, motivated by the real-world applications of CBO, but for now we think this is out of the scope of the current work which is focused on algorithmic developments, providing guarantees, and a reasonable empirical validation.

---

### Author Response · Authors · 2022-11-11
**Author's response to all reviewers and the AC**

We sincerely thank the reviewers for their comments. We have submitted an updated paper based on the reviewer feedback as follows:
- Updated related work: Added more detail about Astudillo and Frazier (2021),  Kusakawa et al. (2021) and grey-box optimization.
- Added appendix A.2.4 which details how $\beta_T$ in the main theorem might scale with $T$.
- Updated equations 7 and 8 with GPs for vector-valued functions and added appendix A.1 for background and guide through the notation.
- Reworked the setup to make the notation and difference between soft and hard interventions clearer.
- Added details on the neural network architecture used for $\eta_i$ in appendix A.3.
- Figure 2 (the illustrations of the DAGs used in the experiments) is moved to the appendix.

Importantly, we noticed that both reviewer geFQ and reviewer wHmj had questions about the similarity of the setup to CBO and function network BO (FNBO) respectively. We want to emphasize that neither setup is totally equivalent to ours and MCBO aims to connect both. Unlike CBO, we allow for soft interventions. Unlike FNBO, we use a causal model formalism that allows for hard interventions and incorporates system noise. Our setup is general enough that it includes both the original CBO setup (assuming no unobserved confounders) and can be applied to FNBO. We believe that an approach applicable in the original CBO setting and to FNBO allows for exchange between these two previously separate ideas (in addition to the other contributions of the paper: new algorithm, improved guarantees, strong empirical performance). Our notation and diagrams reflect the relevance of our work to both the original CBO and FNBO settings.

We are open to provide any further clarification.

---

### Comment · Area_Chair_RKqL · 2022-12-12
**Discussion of Kasakawa et al.**

I'd like to thank the reviewers for their reviews and thank the authors for responding thoughtfully and clearly, and also for uploading an updated version of their paper responding to many of the reviewers' concerns.

I'd like to add one technical point after reading the updated version of the paper. While the paper is much-improved in terms of its discussion of the literature, its discussion of Kusakawa et al. (2021) has two important directions to improve further.

The paper under review writes about Kusakawa et al.,

*Kusakawa et al. (2021) study stochastic function networks in the special case of a chain graph. Similar to MCBO, they develop
a UCB-based acquisition function but use a different approach for computing confidence bounds. Since a UCB acquisition function is not the main focus of Kusakawa et al. (2021), there is no empirical study of this method.*

Based on the arxiv version of Kusakawa et al., paper, https://arxiv.org/pdf/2111.08330.pdf, I see two opportunities for improvement in the current discussion:

- Kusakawa et al. provides a cumulative regret guarantee for one of their UCB-based acquisition function. This appears in the appendix in Theorem B.4. It is important to state this, given that the current paper claims, "We bound [MCBO's] cumulative regret, and obtain the first
non-asymptotic bounds for CBO." It appears that the bound in Kasakawa et al. is non-asymptotic and, while it only applies to cascade-type function networks, it is nevertheless a non-asymptotic bound on cumulative regret for one class of CBO problems. In light of this, it is important to ensure that all claims made about providing the first cumulative regret guarantee of a particular type are clear and correct. If Kusakawa's regret guarantee is discussed in a way that is accurate and makes readers aware of it, then I think it would be appropriate to claim that the current submission is the first to provide a non-asymptotic cumulative regret guarantee for general CBO problems. Another aspect of Kusakawa's regret guarantee is that it appears to depend on a hard-to-quantify information term gamma_T.

- Regarding the statement, "*Since a UCB acquisition function is not the main focus of Kusakawa et al. (2021), there is no empirical study of this method.*" --- Kusakawa et al. (2021) introduces two UCB-based acquisition function. One, called "CI-based", is introduced in the main paper (a simple regret guarantee is provided in Theorem 3.3) and numerical experiments are presented for this algorithm. So, while no empirical study is presented for the UCB-based algorithm introduced in the appendix, there is an empirical study for one of the UCB acquisition functions presented.

---

> ### Author Response · Authors · 2022-12-12
> **Re: Discussion of Kasakawa et al.**
>
> Given your comments, we think an appropriate wording for the quoted section of the related work would be as follows (written here since in this discussion phase I believe we cannot update the submission until the camera-ready).
>
> Kusakawa et al. (2021) study stochastic function networks in the special case of a chain graph. Similar to MCBO, they develop a UCB-based acquisition function and provide an accompanying cumulative regret guarantee but do not employ a reparameterization trick to optimize the acquisition function. Kusakawa et al. do not perform any empirical study of their UCB-based method, and instead evaluate  expected improvement and credible interval methods also developed for the chain graph setting.
>
> Note that we have highlighted the existence of a cumulative regret bound for Kusakawa et al. and given reference to the other methods they study.
>
> Please let us know if this clarifies your concern; we are happy to discuss further.

---

### Decision · Program_Chairs · 2023-01-20

**Decision:**

Accept: notable-top-25%

**Justification For Why Not Higher Score:**

- Experiments are not "exciting" enough for an oral, as articulated by the reviewers
- Level of novelty is not high enough for an oral

**Justification For Why Not Lower Score:**

- Solid analysis and algorithm
- Good improvement over existing methods

**Metareview: Summary, Strengths And Weaknesses:**

This paper studies Causal Bayesian optimization (CBO) in a model that combines models in two lines of literature, one from Aglietti et al. (2020b) on causal BayesOpt that considered "hard interventions", and the other from Astudillo & Frazier (2021b) on BO for function networks that considered "soft interventions".  It proposes a novel acquisition function and proves an associated regret guarantee.  This is the first algorithm for CBO with a cumulative regret guarantee. The algorithm provides better average reward (a surrogate for cumulative regret) than benchmarks in numerical experiments.

Strengths

- Model is general and fills a gap in the literature between past work on CBO and function networks
- Non-trivial acquisition function and cumulative regret guarantee
- Sufficient novelty relative to Aglietti et al., Astudillo & Frazier, and Kusakawa et al.
- Well-written

Weaknesses

- Paper assumes no unobserved confounding, which is unrealistic in many causal inference applications

- The paper's novelty comes from filling a gap between Aglietti et al. 2020, Frazier & Astudillo 2021, and Kasakawa et al. 2021. There is enough novelty for publication, but the level of novelty is smaller than in some other accepted ICLR papers.

- I was surprised that the proposed method did not outperform the benchmarks by more, given that EIFN is designed to minimize simple regret, and UCB is unable to take advantage of problem structure. This may suggest that there is room to further improve the acquisition function.

Originally there were concerns about missing discussion of related literature in the submitted the manuscript including how Kasakawa et al. 2021 was discussed. These were addressed during the review process through conversations with the authors, who proposed edits to the manuscript responding to these concerns.

**Note From Pc:**

if the above contains the word "oral" or "spotlight" please see: "oral" presentation means -> notable-top-5% and "spotlight" means -> notable-top-25%. As stated in our emails, we are disassociating presentation type from AC recommendations